# Data Augmentation Can Improve Robustness

**Sylvestre-Alvise Rebuffi\*, Sven Gowal\*, Dan Calian,**
**Florian Stimberg**, **Olivia Wiles** and **Timothy Mann**
DeepMind, London
{sylvestre,sgowal}@deepmind.com

## Abstract

Adversarial training suffers from *robust overfitting*, a phenomenon where the robust test accuracy starts to decrease during training. In this paper, we focus on reducing robust overfitting by using common data augmentation schemes. We demonstrate that, contrary to previous findings, when combined with model weight averaging, data augmentation can significantly boost robust accuracy. Furthermore, we compare various data augmentations techniques and observe that spatial composition techniques work best for adversarial training. Finally, we evaluate our approach on CIFAR-10 against $\ell_\infty$ and $\ell_2$ norm-bounded perturbations of size $\epsilon = 8/255$ and $\epsilon = 128/255$, respectively. We show large absolute improvements of +2.93% and +2.16% in robust accuracy compared to previous state-of-the-art methods. In particular, against $\ell_\infty$ norm-bounded perturbations of size $\epsilon = 8/255$, our model reaches 60.07% robust accuracy without using any external data. We also achieve a significant performance boost with this approach while using other architectures and datasets such as CIFAR-100, SVHN and TINYIMAGENET.

## 1 Introduction

Despite their success, neural networks are not intrinsically robust. In particular, it has been shown that the addition of imperceptible deviations to the input, called adversarial perturbations, can cause neural networks to make incorrect predictions with high confidence [5, 6, 18, 32, 49]. Starting with Szegedy et al. [49], there has been a lot of work on understanding and generating adversarial perturbations [2, 6], and on building defenses that are robust to such perturbations [18, 29, 34, 41]. Unfortunately, many of the defenses proposed in the literature target failure cases found through specific adversaries, and as such they are easily broken by different adversaries [3, 53]. Among successful defenses are robust optimization techniques like the one by Madry et al. [34] that learns robust models by finding worst-case adversarial perturbations at each training step before adding them to the training data. In fact, adversarial training as proposed by Madry et al. is so effective [20] that it is the de facto standard for training adversarially robust neural networks. Indeed, since Madry et al. [34], various modifications to their original implementation have been proposed [20, 27, 40, 44, 57, 63].

Notably, Carmon et al. [7], Hendrycks et al. [25], Najafi et al. [36], Uesato et al. [54], Zhai et al. [60] showed that using additional data improves adversarial robustness, while Gowal et al. [20], Rice et al. [44], Wu et al. [56] found that data augmentation techniques did not boost robustness. This dichotomy motivates this paper. In particular, we explore whether it is possible to fix the training procedure such that data augmentation becomes useful in the setting without additional data. By making the observation that model weight averaging (WA) [28] helps robust generalization to a wider extent when robust overfitting is minimized, we propose to combine model weight averaging with data augmentation techniques. Overall, we make the following contributions:

- We demonstrate that, when combined with model weight averaging, data augmentation techniques such as *Cutout* [15], *CutMix* [58] and *MixUp* [62] can improve robustness.

35th Conference on Neural Information Processing Systems (NeurIPS 2021).

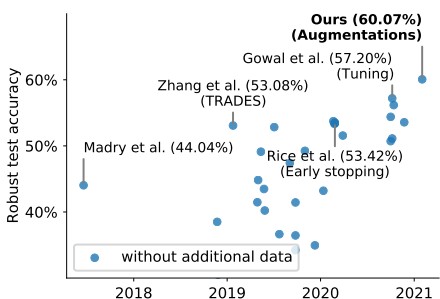

Figure 1: Robust accuracy of various models submitted to RobustBench [11] against AUTOATTACK [10] on CIFAR-10 with $\ell_\infty$ perturbations of size $8/255$ displayed in publication order. Our method builds on Gowal et al. [20] (shown above with 57.20%) and explores how augmented data can be used to improve robust accuracy by +2.87% without using any additional external data.

- To the contrary of Gowal et al. [20], Rice et al. [44], Wu et al. [56] which all tried data augmentation techniques without success, we are able to use any of these three aforementioned techniques to obtain new state-of-the-art robust accuracies (see Figure 1). We find *CutMix* to be the most effective method by reaching a robust accuracy of 60.07% on CIFAR-10 against $\ell_\infty$ perturbations of size $\epsilon = 8/255$ (an improvement of +2.93% upon the state-of-the-art).

- We conduct thorough experiments to show that our approach generalizes across architectures, datasets and threat models. We also investigate the trade-off between robust overfitting and underfitting to explain why *MixUp* performs worse than spatial composition techniques.

- Finally, we provide empirical evidence that weight averaging exploits data augmentation by ensembling model snapshots which have the same total accuracy but differ at the individual prediction level.

## 2 Related Work

**Adversarial $\ell_p$-norm attacks.** Since Szegedy et al. [49] observed that neural networks which achieve high accuracy on test data are highly vulnerable to adversarial examples, the art of crafting increasingly sophisticated adversarial examples has received a lot of attention. Goodfellow et al. [18] proposed the Fast Gradient Sign Method (FGSM) which generates adversarial examples with a single normalized gradient step. It was followed by R+FGSM [52], which adds a randomization step, and the Basic Iterative Method (BIM) [32], which takes multiple smaller gradient steps.

**Adversarial training as a defense.** The adversarial training procedure [34] feeds adversarially perturbed examples back into the training data. It is widely regarded as one of the most successful method to train robust deep neural networks. It has been augmented in different ways – with changes in the attack procedure (e.g., by incorporating momentum [16]), loss function (e.g., logit pairing [35]) or model architecture (e.g., feature denoising [57]). Another notable work by Zhang et al. [63] proposed TRADES, which balances the trade-off between standard and robust accuracy, and achieved state-of-the-art performance against $\ell_\infty$ norm-bounded perturbations on CIFAR-10. More recently, the work from Rice et al. [44] studied *robust overfitting* and demonstrated that improvements similar to TRADES could be obtained more easily using classical adversarial training with early stopping. This later study revealed that early stopping was competitive with many other regularization techniques and demonstrated that data augmentation schemes beyond the typical *random padding-and-cropping* were ineffective on CIFAR-10. Finally, Gowal et al. [20] highlighted how different hyper-parameters (such as network size and model weight averaging) affect robustness. They were able to obtain models that significantly improved upon the state-of-the-art, but lacked a thorough investigation on data augmentation schemes. Similarly to Rice et al. [44], they also make the conclusion that data augmentations beyond *random padding-and-cropping* do not improve robustness.

**Data augmentation.** Data augmentation has been shown to improve the generalisation of standard (non-robust) training. For image classification tasks, random flips, rotations and crops are commonly used [23]. More sophisticated techniques such as *Cutout* [15] (which produces random occlusions), *CutMix* [58] (which replaces parts of an image with another) and *MixUp* [62] (which linearly interpolates between two images) all demonstrate extremely compelling results. As such, it is rather surprising that they remain ineffective when training adversarially robust networks [20, 44, 56]. In this work, we revisit these common augmentation techniques in the context of adversarial training.

# 3 Preliminaries and hypothesis

The rest of this manuscript is organized as follows. In this section, we provide an overview of adversarial training and introduce the hypothesis that model weight averaging works better when robust overfitting is reduced. In section 4 we discuss that data augmentations can be used to verify this hypothesis. Finally, we provide thorough experimental results in section 6.

## 3.1 Adversarial training

Madry et al. [34] formulate a saddle point problem to find model parameters $\boldsymbol{\theta}$ that minimize the adversarial risk:

$$\arg\min_{\boldsymbol{\theta}} \mathbb{E}_{(\boldsymbol{x},y)\sim\mathcal{D}} \left[ \max_{\boldsymbol{\delta}\in\mathbb{S}} l(f(\boldsymbol{x}+\boldsymbol{\delta};\boldsymbol{\theta}),y) \right] \tag{1}$$

where $\mathcal{D}$ is a data distribution over pairs of examples $\boldsymbol{x}$ and corresponding labels $y$, $f(\cdot;\boldsymbol{\theta})$ is a model parametrized by $\boldsymbol{\theta}$, $l$ is a suitable loss function (such as the $0-1$ loss in the context of classification tasks), and $\mathbb{S}$ defines the set of allowed perturbations. For $\ell_p$ norm-bounded perturbations of size $\epsilon$, the adversarial set is defined as $\mathbb{S}_p = \{\boldsymbol{\delta} \mid \|\boldsymbol{\delta}\|_p \leq \epsilon\}$. In the rest of this manuscript, we will use $\epsilon_p$ to denote $\ell_p$ norm-bounded perturbations of size $\epsilon$ (e.g., $\epsilon_\infty = 8/255$). To solve the inner optimization problem, Madry et al. [34] use Projected Gradient Descent (PGD), which replaces the non-differentiable $0-1$ loss $l$ with the cross-entropy loss $l_{\text{ce}}$ and computes an adversarial perturbation $\hat{\boldsymbol{\delta}} = \boldsymbol{\delta}^{(K)}$ in $K$ gradient ascent steps of size $\alpha$ as

$$\boldsymbol{\delta}^{(k+1)} \leftarrow \text{proj}_{\mathbb{S}} \left( \boldsymbol{\delta}^{(k)} + \alpha\,\text{sign}\left( \nabla_{\boldsymbol{\delta}^{(k)}} l_{\text{ce}}(f(\boldsymbol{x}+\boldsymbol{\delta}^{(k)};\boldsymbol{\theta}),y) \right) \right) \tag{2}$$

where $\boldsymbol{\delta}^{(0)}$ is chosen at random within $\mathbb{S}$, and where $\text{proj}_{\mathbb{A}}(\boldsymbol{a})$ projects a point $\boldsymbol{a}$ back onto a set $\mathbb{A}$, $\text{proj}_{\mathbb{A}}(\boldsymbol{a}) = \text{argmin}_{\boldsymbol{a}'\in\mathbb{A}}\|\boldsymbol{a}-\boldsymbol{a}'\|_2$. We refer to this inner optimization with $K$ steps as $\text{PGD}^K$.

## 3.2 Robust overfitting

To the contrary of standard training, which often shows no *overfitting* in practice [61], adversarial training suffers from *robust overfitting* [44]. Robust overfitting is the phenomenon by which robust accuracy on the test set quickly degrades while it continues to rise on the train set (clean accuracy on both sets continues to improve as well). Rice et al. [44] propose to use early stopping as the main contingency against robust overfitting, and demonstrate that it also allows to train models that are more robust than those trained with other regularization techniques (such as data augmentation or increased $\ell_2$-regularization). They observed that some of these other regularization techniques could reduce the impact of overfitting at the cost of producing models that are over-regularized and lack overall robustness and accuracy. There is one notable exception which is the addition of external data [7, 54]. Figure 2(a) shows how the robust accuracy (evaluated on the test set) evolves as training progresses on CIFAR-10 against $\epsilon_\infty = 8/255$. Without external data, robust overfitting is clearly visible and appears shortly after the learning rate is dropped (the learning rate is decayed by $10\times$ two-thirds through training in a schedule similar to [44] and commonly used since [34]). Robust overfitting completely disappears when an additional set of 500K pseudo-labeled images (see Carmon et al. [7]) is introduced.

## 3.3 Model weight averaging

Model weight averaging (WA) [28] can be implemented using an exponential moving average $\boldsymbol{\theta}'$ of the model parameters $\boldsymbol{\theta}$ with a decay rate $\tau$ (i.e., $\boldsymbol{\theta}' \leftarrow \tau \cdot \boldsymbol{\theta}' + (1-\tau) \cdot \boldsymbol{\theta}$ at each training step). During evaluation, the weighted parameters $\boldsymbol{\theta}'$ are used instead of the trained parameters $\boldsymbol{\theta}$. Chen et al. [8], Gowal et al. [20] discovered that model weight averaging can significantly improve robustness on a wide range of models and datasets. Chen et al. [8] argue (similarly to [56]) that WA leads to a flatter adversarial loss landscape, and thus a smaller robust generalization gap. Gowal et al. [20] also explain that, in addition to improved robustness, WA reduces sensitivity to early stopping. While this is true, it is important to note that WA is still prone to robust overfitting. This is not surprising, since the exponential moving average "forgets" older model parameters as training goes on. Figure 2(b) shows how the robust accuracy evolves as training progresses when using WA. We observe that, after the change of learning rate, the averaged weights are increasingly affected by overfitting, thus resulting in worse robust accuracy for the averaged model.

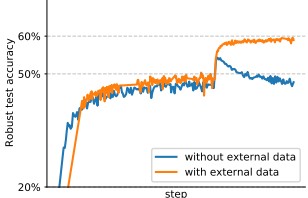 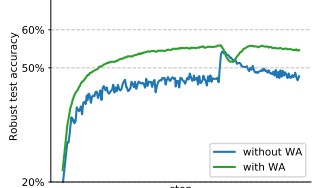 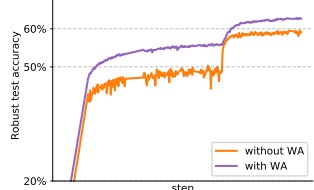

(a) Adversarial training with and without additional data from 80M-Tɪ (without WA)

(b) Effect of WA without external data

(c) Effect of WA with external data

Figure 2: We compare the robust accuracy against $\epsilon_\infty = 8/255$ on Cɪꜰᴀʀ-10 of an adversarially trained Wide ResNet (Wʀɴ)-28-10. Panel (a) shows the impact of using additional external data from 80M-Tɪ [51] and illustrates *robust overfitting*. Panel (b) shows the benefit of *model weight averaging* (WA) despite robust overfitting. Panel (c) shows that WA remains effective and useful even when robust overfitting disappears. The graphs show the evolution of the robust accuracy as training progresses (against Pɢᴅ⁴⁰). The jump two-thirds through training is due to a drop in learning rate.

## 3.4 Hypothesis

As WA results in flatter, wider solutions compared to the steep decrease in robust accuracy observed for Stochastic Gradient Descent (SGD) [8], it is natural to ask ourselves whether WA remains useful in cases that do not exhibit robust overfitting. Figure 2(c) shows how the robust accuracy evolves as training progresses when using WA and additional external data (for which standard SGD does not show signs of overfitting). We notice that the robust performance in this setting is not only preserved but even boosted when using WA. Hence, we formulate the hypothesis that model weight averaging helps robustness to a greater extent when robust accuracy between model iterations can be maintained. This hypothesis is also motivated by the observation that WA acts as a temporal ensemble – akin to Fast Geometric Ensembling by Garipov et al. [17] who show that efficient ensembling can be obtained by aggregating multiple checkpoint parameters at different training times. As such, to improve robustness, it is important to ensemble a suite of equally strong and diverse models. Although mildly successful, we note that ensembling has received some attention in the context of adversarial training [39, 48]. In particular, Grefenstette et al. [22], Tramèr et al. [52] found that ensembling could reduce the risk of gradient obfuscation caused by locally non-linear loss surfaces.

## 4 Data augmentations

**Limiting robust overfitting without external data.** Rice et al. [44] show that combining data augmentation methods such as *Cutout* or *MixUp* with early stopping does not improve robustness upon early stopping alone. While, these methods do not improve upon the "best" robust accuracy, they reduce the extent of robust overfitting, thus resulting in a slower decrease in robust accuracy compared to classical adversarial training (which uses random crops and weight decay). This can be seen in Figure 3(a) where *MixUp* without WA exhibits no decrease in robust accuracy, whereas the robust accuracy of the standard combination of *random padding-and-cropping* without WA (*Pad & Crop*) decreases immediately after the change of learning rate.

**Testing the hypothesis.** Since *MixUp* preserves robust accuracy while *Pad & Crop* does not, this comparison can be used to evaluate the hypothesis that WA is more beneficial when the performance between model iterations is maintained. Therefore, we compare in Figure 3(b) the effect of WA on robustness when using *MixUp*. We observe that, when using WA, the performance of *MixUp* surpasses the performance of *Pad & Crop*. Indeed, the robust accuracy obtained by the averaged weights of *Pad & Crop* (in blue) slowly decreases after the change of learning rate, while the one obtained by *MixUp* (in green) increases throughout training[1]. Ultimately, *MixUp* with WA obtains a higher robust accuracy despite the fact that the non-averaged *MixUp* model has a significantly lower "best" robust accuracy than the non-averaged *Pad & Crop* model. This finding is notable as it

---

[1]The accuracy drop just after the change of learning rate stems from averaging very different weights.

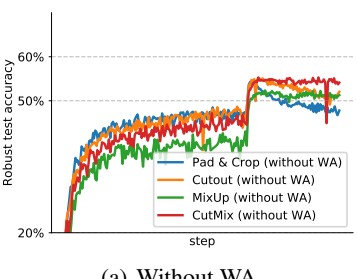

(a) Without WA

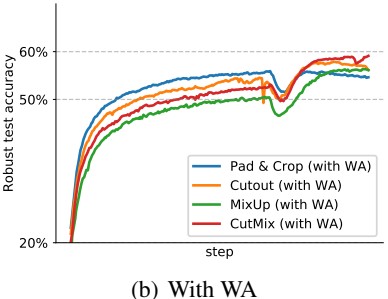

(b) With WA

Figure 3: Accuracy against $\epsilon_\infty = 8/255$ on CIFAR-10 with and without using model weight averaging (WA) for different data augmentation schemes. The model is a WRN-28-10 and both panels show the evolution of the robust accuracy as training progresses (against PGD$^{40}$). The jump in robust accuracy two-thirds through training is due to a drop in learning rate.

demonstrates for the first time the benefits of data augmentation schemes for adversarial training (this contradicts the findings from three recent publications: [20, 44, 56]).

**Exploring data augmentations.** After verifying our hypothesis for *MixUp*, we investigate if other augmentations can help maintain robust accuracy and also be combined with WA to improve robustness. We concentrate on the following image patching techniques: *Cutout* [15] which inserts empty image patches and *CutMix* [58] which replaces part of an image with another. In section 6, we also evaluate *RICAP* [50] and *SmoothMix* [33]. We describe more thoroughly these augmentations in Appendix B where we also study additional augmentations with *AutoAugment* [12] and *RandAugment* [13]. Similarly to the analysis done for *MixUp*, we report in Figure 3 the robust accuracy obtained by *Cutout* and *CutMix* with and without WA throughout training. First, we note that these two techniques achieve a higher "best" robust accuracy than *MixUp*, as shown in Figure 3(a). The "best" robust accuracy obtained by *Cutout* and *CutMix* is roughly identical to the one obtained by *Pad & Crop*, which is consistent with the results from Rice et al. [44]. Second, while *Cutout* suffers from robust overfitting, *CutMix* does not. Hence, as demonstrated in the previous sections, we expect WA to be more useful with *CutMix*. Indeed, we observe in Figure 3(b) that the robust accuracy of the averaged model trained with *CutMix* keeps increasing throughout training and that its maximum value is significantly above the best accuracy reached by the other augmentation methods. In section 6, we conduct thorough evaluations of these methods against stronger attacks.

## 5 Experimental setup

**Architecture.** We use WRNs [23, 59] as our backbone network. This is consistent with prior work [20, 34, 44, 54, 63] which use diverse variants of this network family. Furthermore, we adopt the same architecture details as Gowal et al. [20] with Swish/SiLU [24] activation functions. Most of the experiments are conducted on a WRN-28-10 model which has a depth of 28, a width multiplier of 10 and contains 36M parameters. To evaluate the effect of data augmentations on wider and deeper networks, we also run several experiments using WRN-70-16, which contains 267M parameters.

**Outer minimization.** We use TRADES [63] optimized using SGD with Nesterov momentum [37, 42] and a global weight decay of $5 \times 10^{-4}$. We train for 400 epochs with a batch size of 512 split over 32 Google Cloud TPU v3 cores [4], and the learning rate is initially set to 0.1 and decayed by a factor 10 two-thirds-of-the-way through training. We scale the learning rates using the linear scaling rule of Goyal et al. [21] (i.e., effective LR $= \max(\text{LR} \times \text{batch size}/256, \text{LR})$). The decay rate of WA is set to $\tau = 0.999$. With these settings, training a WRN-28-10 takes on average 2.5 hours.

**Inner minimization.** Adversarial examples are obtained by maximizing the Kullback-Leibler divergence between the predictions made on clean inputs and those made on adversarial inputs [63]. This optimization procedure is done using the Adam optimizer [30] for 10 PGD steps. We take an initial step-size of 0.1 which is then decreased to 0.01 after 5 steps.

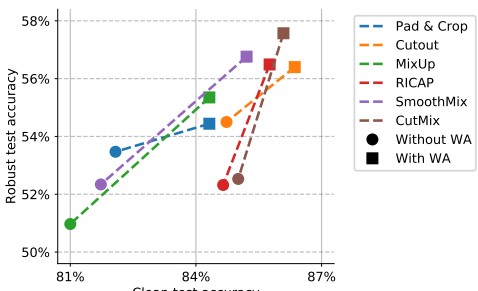
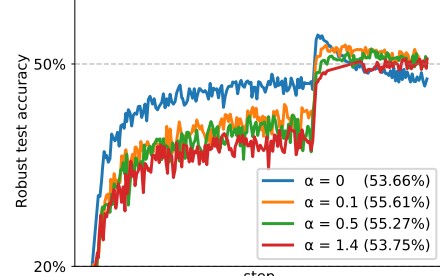

Figure 4: Clean (without adversarial attacks) accuracy and robust accuracy (against AA+MT) for a WRN-28-10 trained against $\epsilon_\infty = 8/255$ on CIFAR-10 for different data augmentation techniques. The lines from circles to squares represent the performance change obtained when using WA.

Figure 5: The graph shows the robust test accuracy against PGD$^{40}$ with $\epsilon_\infty = 8/255$ on CIFAR-10 without using WA as we vary the mixing rate $\alpha$ of *MixUp*. We report in the legend the robust accuracy (against AA+MT) after applying weight averaging to the corresponding runs.

**Evaluation.** We follow the evaluation protocol designed by Gowal et al. [20]. Specifically, we train two (and only two) models for each hyperparameter setting, perform early stopping for each model on a separate validation set of 1024 samples using PGD$^{40}$ similarly to Rice et al. [44] and pick the best model by evaluating the robust accuracy on the same validation set . Finally, we report the robust test accuracy against a mixture of AUTOATTACK [10] and MULTITARGETED [19], which is denoted by AA+MT. This mixture consists in completing the following sequence of attacks: AUTOPGD on the cross-entropy loss with 5 restarts and 100 steps, AUTOPGD on the difference of logits ratio loss with 5 restarts and 100 steps and finally MULTITARGETED on the margin loss with 10 restarts and 200 steps. The training curves, such as those visible in Figure 2, are always computed using PGD with 40 steps and the Adam optimizer (with step-size decayed by $10\times$ at step 20 and 30).

## 6 Experimental results

First, we will investigate which augmentation techniques benefit the most from WA and why. Then, we will generalize our approach to other architecture, threat model and datasets. Finally, we provide empirical evidence showing that WA exploits data augmentation by ensembling model snapshots which differ at the individual prediction level.

### 6.1 Comparing data augmentations

Here, we compare data augmentations with and without WA. We consider as baseline the *Pad & Crop* augmentation which reproduces the current state-of-the-art set by Gowal et al. [20]. This augmentation consists in first padding the image by 4 pixels on each side and then taking a random $32 \times 32$ crop. In Figure 4, we compare this baseline with various data augmentations, *MixUp*, *Cutout*, *CutMix*, *RICAP* and *SmoothMix*. A first cluster (the four top squares), containing *RICAP*, *Cutout*, *SmoothMix* and *CutMix*, includes the four methods that occlude local information with patching and provide a significant boost upon the baseline with +3.06% in robust accuracy for *CutMix* and an average improvement of +1.54% in clean accuracy. The other cluster, with *MixUp*, only improves the robust accuracy upon the baseline by a small margin of +0.91%. Furthermore, we also point out that *Pad & Crop* and *Cutout*, which were the two augmentations suffering from robust overfitting in Figure 3(a), benefit the least of WA in Figure 4 (smaller vertical gains). This is consistent with our hypothesis of section 3 that WA is the most beneficial when robust overfitting is reduced.

***MixUp*.** A possible explanation to the worse performance of *MixUp* lies in the fact that *MixUp*, which samples the image mixing weight with a beta distribution $\text{Beta}(\alpha, \alpha)$, tends to either produce images that are far from the original data distribution (when $\alpha$ is large) or too close to the original samples (when $\alpha$ is small). In fact, Figure 5, which shows the robust accuracy when training without WA, illustrates the trade-off between robust overfitting and underfitting as increasing $\alpha$ can lead

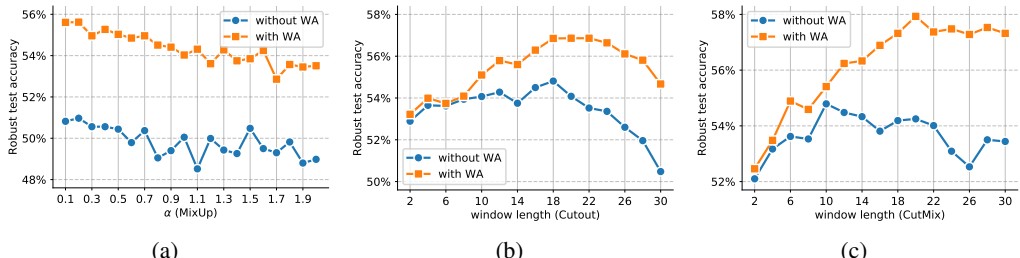

(a)               (b)               (c)

Figure 6: Robust test accuracy against AA+MT with $\epsilon_\infty = 8/255$ on CIFAR-10 as we vary (a) the mixing rate $\alpha$ of *MixUp*, (b) the window length when using *Cutout* and (c) the window length when using *CutMix*. The model is a WRN-28-10 and we compare the settings without and with WA. As a reference when training only with *Pad & Crop*, the same model with WA and without WA reaches 54.44% and 53.66% robust accuracy, respectively. Similarly, without any augmentation, the models with WA and without WA achieve 49.74% and 42.27%, respectively.

to robust underfitting (red curve) while an $\alpha$ too close to 0 would lead to robust overfitting. More specifically, we show in Figure 6(a) that *MixUp*'s robust accuracy against AA+MT keeps decreasing as $\alpha$ increases and the best performance with WA is reached at $\alpha = 0.2$, corresponding to blended images close to the original images.

**Spatial composition techniques.** Figure 6(b,c) show the robust test accuracy as we vary the window length of the patches when using *Cutout* and *CutMix*[2]. We observe that these two techniques are the most beneficial when using large window lengths with a peak reached at a length of 20. Hence, contrary to *MixUp*, they work best with patched images which greatly differ from the original images. This performance gap between *MixUp* and *Cutout*/*CutMix* illustrates a notable difference in the use of data augmentation for adversarial training compared to nominal training. Indeed, adversarial training can lead to underfitting. This leads to some augmentation techniques working better than others in the context of adversarial training. This is the case for spatial composition techniques which outperform blending techniques like *MixUp*. A possible explanation is that low-level features tend to be destroyed by *MixUp*, whereas composition techniques locally maintain these low-level features. Hence, we hypothesize that augmentations designed for robustness need to preserve low-level features. We provide further evidence to support this hypothesis in Appendix B by showing that some components of *RandAugment* such as *Posterize* or *Invert* are detrimental to adversarial robustness.

## 6.2 Generalizing to other architectures, threat model and datasets

**Generalizing to other architectures.** Table 1 shows the performance of *CutMix* and the *Pad & Crop* baseline when varying the model architecture and size. We experiment with different variants of WideResNet and ResNet. We use WA for both *CutMix* and the *Pad & Crop* baseline. We observe that *CutMix* consistently outperforms the baseline by at least +2.90% in robust accuracy across all model sizes for WideResNet and by at least +1.76% for ResNet models.

**Generalizing to another threat model.** We extend our evaluation to $\ell_2$-norm bounded perturbations. Table 2 shows the performance of data augmentation on CIFAR-10 against $\epsilon_\infty = 8/255$ and $\epsilon_2 = 128/255$. We observe that using *CutMix* provides a significant boost in robust accuracy for both threat models with up to +2.93% (in the $\ell_\infty$ setting) and +2.16% (in the $\ell_2$ setting).

**Generalizing to other datasets.** To evaluate the generality of our approach, we evaluate it on CIFAR-100 [31], SVHN [38] and TINYIMAGENET [45] and we report the results in Table 3. First, on CIFAR-100, our best model reaches 32.43% against AUTOATTACK and improves noticeably on the state-of-the-art by +2.40% (in the setting that does not use any external data). Second, on TINYIMAGENET with a WRN-28-10 we obtain a significant +2.00% boost for robust accuracy against

---

[2]For all CutMix results except those reported in Fig 6(c)), the window length is randomly sampled such that the patch area ratio follows a beta distribution with parameters $\alpha = 1$ and $\beta = 1$.

| Setup | Pad & Crop | | CutMix | |
|---|---|---|---|---|
| | Clean | Robust | Clean | Robust |
| **Varying the architecture** | | | | |
| ResNet-18 | 83.12% | 50.52% | 80.57% | **52.28%** |
| ResNet-34 | 84.68% | 52.52% | 83.35% | **54.80%** |
| WRN-28-10 | 84.32% | 54.44% | 86.09% | **57.50%** |
| WRN-34-10 | 84.89% | 55.13% | 86.18% | **58.09%** |
| WRN-34-20 | 85.80% | 55.69% | 87.80% | **59.25%** |
| WRN-70-16 | 86.02% | 57.17% | 87.25% | **60.07%** |

Table 1: Robust test accuracy (against AA+MT) against $\epsilon_\infty = 8/255$ on CIFAR-10 for different architectures. In all cases, we use weight averaging and we compare *Pad & Crop* and *CutMix*.

| Setup | $\ell_\infty$ | | $\ell_2$ | |
|---|---|---|---|---|
| | Clean | Robust | Clean | Robust |
| **WRN-28-10** | | | | |
| Gowal et al. [20] (trained by us) | 84.32% | 54.44% | 88.60% | 72.56% |
| Ours (CutMix) | 86.22% | **57.50%** | 91.35% | **76.12%** |
| **WRN-70-16** | | | | |
| Gowal et al. [20] (trained by us) | 85.29% | 57.14% | 90.90% | 74.50% |
| Ours (CutMix) | 87.25% | **60.07%** | 92.43% | **76.66%** |

Table 2: Clean (without adversarial attacks) accuracy and robust accuracy (against AA+MT) on CIFAR-10 as we both test against $\epsilon_\infty = 8/255$ and $\epsilon_2 = 128/255$.

| Model | Clean | AA+MT | AA |
|---|---|---|---|
| **CIFAR-100** | | | |
| Cui et al. [14] (WRN-34-10) | 60.64% | – | 29.33% |
| WRN-28-10 (retrained) | 59.05% | 28.75% | – |
| WRN-28-10 (CutMix) | 62.97% | **30.50%** | **29.80%** |
| Gowal et al. [20] (WRN-70-16) | 60.86% | 30.67% | 30.03% |
| WRN-70-16 (retrained) | 59.65% | 30.62% | – |
| WRN-70-16 (CutMix) | 65.76% | **33.24%** | **32.43%** |
| **SVHN** | | | |
| WRN-28-10 (retrained) | 92.87% | 56.83% | – |
| WRN-28-10 (CutMix) | 94.52% | **57.32%** | – |
| **TINYIMAGENET** | | | |
| WRN-28-10 (retrained) | 53.27% | 21.83% | – |
| WRN-28-10 (CutMix) | 53.69% | **23.83%** | – |

Table 3: Clean and robust accuracy (AA+MT and AUTOATTACK for select models) on CIFAR-100, SVHN and TINY-IMAGENET against $\epsilon_\infty = 8/255$ obtained by different models (with WA). The 're-trained' indication means that the models have been retrained according to Gowal et al. [20]'s methodology.

AA+MT with $\epsilon_\infty = 8/255$. Finally, on SVHN, our best model reaches 57.32% against AA+MT and improves on the baseline by a smaller margin than on CIFAR-10, CIFAR-100 or TINYIMAGENET. This smaller improvement is expected as *CutMix* is not suited to SVHN because images of SVHN contain multiple digits per image.

### 6.3 Empirical elements on how weight averaging exploits data augmentation

**Motivating model ensembling.** First, we show that model ensembling can be used to improve robust accuracy. To do so, we evaluate ensembling early-stopped models which have been trained from scratch independently. We ensemble two early-stopped WRN-28-10 models trained on CIFAR-10 with *Pad & Crop* by taking the average of the two independent models at the prediction level. In spite of this naive ensembling approach, we observe a significant boost in robust performance as this ensemble reaches 55.69% robust accuracy against AA+MT compared to 54.44% with a single model. Hence, even a simple ensemble of two independent runs can exploit the variance in individual robust predictions. Actually, the boost in robust accuracy is even stronger when ensembling two early-stopped WRN-28-10 models trained with *CutMix* as the ensemble reaches 56.35% robust accuracy which is +3.82% better compared to an individual model. Augmentation techniques such as *CutMix* promote more diversity between runs than *Pad & Crop*, leading thereby to better robust performance when ensembling. This is further evidence that ensembling by its ability of exploiting the diversity of the models is mainly responsible for robustness improvements.

**Model ensembling by weight averaging.** We would like to ensemble more than two models but it would be inefficient computationally and memory wise to average the predictions of many independently trained models. To circumvent this issue, the naive ensembling approach is replaced by model weight averaging as we exploit the commonly known fact [8, 19, 43] that models trained with adversarial training tend to be locally linear. Indeed, under the assumption of linearity, weight averaging becomes equivalent to model ensembling. Hence, instead of ensembling independently trained models, weight averaging ensembles model iterations obtained during one training run. As discussed in the previous paragraph, model ensembling improves robustness by exploiting the diversity of equally performing models so we need the model iterations used with weight averaging to have similar robust performance but also some diversity in individual robust predictions. As we

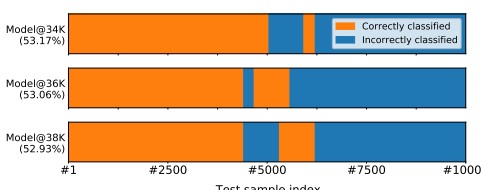

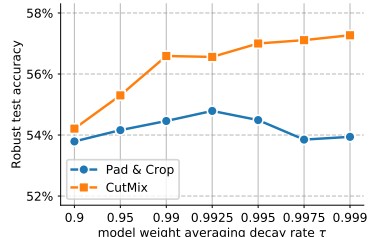

Figure 7: The bar plots show the outcome of each individual robust prediction for different snapshots of a same training run of a WRN-28-10 against $\epsilon_\infty = 8/255$ on CIFAR-10 without model weight averaging. The test sample indices have been re-ordered such as to show contiguous blocks. The plots show a significant variation in individual robust predictions across different snapshots while the total robust accuracy (i.e. the number in parenthesis) remains stable.

Figure 8: Robust test accuracy against AA+MT with $\epsilon_\infty = 8/255$ on CIFAR-10 as we vary the decay rate of the model weight averaging. The model is a WRN-28-10, which is trained either with *CutMix* or *Pad & Crop*.

have previously seen in Figure 3(a), *CutMix* without weight averaging prevents robust overfitting and leads to a flat robust accuracy after the change of learning rate. Hence, these model snapshots share the same total accuracy but we would like to know if they differ at the individual prediction level. Figure 7 represents the individual robust predictions on test samples for three snapshots taken during training. While these three snapshots roughly have the same number of correctly classified samples, we see that there are on average 888 errors (out of 10k samples) per snapshot which are not made in the other two snapshots. This shows that augmentations which avoid robust overfitting such as *CutMix* produce diverse and equally performing model iterations, which can be ensembled effectively by model weight averaging, leading thereby to improved robust performance.

**The limits when exploiting the diversity between model iterations.** When robust overfitting occurs, weight averaging is still helpful but to a lesser extent. In fact, a compromise must be found between the performance boost from ensembling diverse model iterations and the performance loss from incorporating model iterations with degraded performance in the ensemble. We illustrate this point by running an ablation study in Figure 8 measuring the robust accuracy obtained when varying the decay rate $\tau$ of model weight averaging and using either *Pad & Crop* or *CutMix*. While for *CutMix* increasing the weight averaging decay rate (i.e. ensembling more model iterations) always results in better robust performance, we observe that for *Pad & Crop* the maximum robust performance is obtained at $\tau = 0.9925$. When the weight averaging decay rate becomes too large ($\tau > 0.9925$), too many model iterations with degraded performance are incorporated in the ensemble, thus hurting the robust performance of the ensemble. Hence, the diversity between model iterations can only compensate up to a certain point for the decrease in robust performance due to robust overfitting.

## 7 Conclusion

Contrary to previous works [20, 44, 56], which have tried data augmentation techniques to train adversarially robust models without success, we demonstrate that combining data augmentations with model weight averaging can significantly improve robustness. We also provide insights on why weight averaging works better with data augmentations which reduce robust overfitting. We show in fact that model snapshots of a same run have the same total robust accuracy but they greatly differ at the individual prediction level, thus allowing a performance boost when ensembling these snapshots. Code and models are available online at `https://github.com/deepmind/deepmind-research/tree/master/adversarial_robustness`.

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
