# A Analysis of models

In this section, we perform additional diagnostics that give us confidence that our models are not doing any form of gradient obfuscation or masking [3, 53].

**AUTOATTACK and robustness against black-box attacks.** First, we report in Table 4 the robust accuracy obtained by our strongest models against a diverse set of attacks. These attacks are run as a cascade using the AUTOATTACK library available at `https://github.com/fra31/auto-attack`. The cascade is composed as follows:

- AUTOPGD-CE, an untargeted attack using PGD with an adaptive step on the cross-entropy loss [10],
- AUTOPGD-T, a targeted attack using PGD with an adaptive step on the difference of logits ratio [10],
- FAB-T, a targeted attack which minimizes the norm of adversarial perturbations [9],
- SQUARE, a query-efficient black-box attack [1].

First, we observe that our combination of attacks, denoted AA+MT matches the final robust accuracy measured by AUTOATTACK. Second, we also notice that the black-box attack (i.e., SQUARE) does not find any additional adversarial examples. Overall, these results indicate that our empirical measurement of robustness is meaningful and that our models do not obfuscate gradients.

| MODEL | NORM | RADIUS | AUTOPGD-CE | + AUTOPGD-T | + FAB-T | + SQUARE | CLEAN | AA+MT |
|---|---|---|---|---|---|---|---|---|
| WRN-28-10 (CutMix) | | | 61.01% | 57.61% | 57.61% | 57.61% | 86.22% | 57.50% |
| WRN-70-16 (CutMix) | $\ell_\infty$ | $\epsilon = 8/255$ | 62.65% | 60.07% | 60.07% | 60.07% | 87.25% | 60.07% |
| WRN-70-16 [20] | | | 59.39% | 57.21% | 57.20% | 57.20% | 85.29% | 57.14% |

Table 4: Clean (without adversarial attacks) accuracy and robust accuracy (against the different stages of AUTOATTACK) on CIFAR-10 obtained by different models with our method. We compare them with the results of [20] for a WRN-70-16 in the case without additional data. Refer to `https://github.com/fra31/auto-attack` for more details.

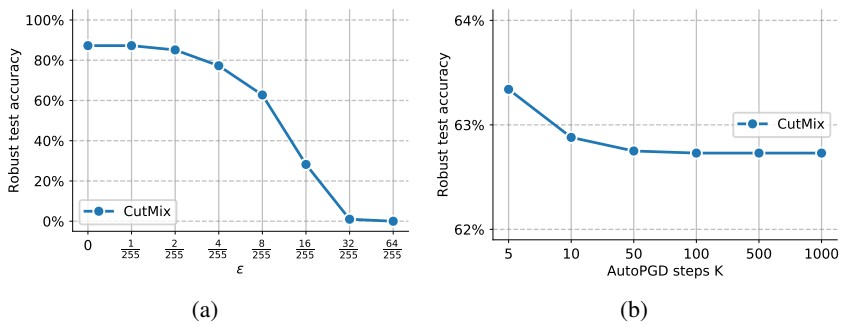

(a)            (b)

Figure 9: Robust test accuracy measured by running AUTOPGD-CE with (a) different radii $\epsilon_\infty$ and (b) different number of steps $K$. The model is a WRN-70-16 network trained with *CutMix* against $\epsilon_\infty = 8/255$, which obtains 60.07% robust accuracy against AA+MT at $\epsilon_\infty = 8/255$.

**Further analysis of gradient obfuscation.** In this paragraph, we consider a WRN-70-16 trained with *CutMix* against $\epsilon_\infty = 8/255$, which obtains 60.07% robust accuracy against AA+MT at $\epsilon_\infty = 8/255$.

In Figure 9(a), we run AUTOPGD-CE with 100 steps and 1 restart and we vary the perturbation radius $\epsilon_\infty$ between zero and $64/255$. As expected, the robust accuracy gradually drops as the radius increases indicating that PGD-based attacks can find adversarial examples and are not hindered by gradient obfuscation.

In Figure 9(b), we run AUTOPGD-CE with $\epsilon_\infty = 8/255$ and 1 restart and vary the number of steps $K$ between five and 1000. We observe that the measured robust accuracy converges after 50 steps. This is further indication that attacks converge in 100 steps.

**Loss landscapes.** Finally, we analyze the adversarial loss landscapes of the model considered in the previous paragraph. To generate a loss landscape, we vary the network input along the linear space defined by the worse perturbation found by $\text{PGD}^{40}$ ($u$ direction) and a random Rademacher direction ($v$ direction). The $u$ and $v$ axes represent the magnitude of the perturbation added in each of these directions respectively and the $z$ axis is the adversarial margin loss [6]: $z_y - \max_{i \neq y} z_i$ (i.e., a misclassification occurs when this value falls below zero).

Figure 10 shows the loss landscapes around the first 2 images of the CIFAR-10 test set. All landscapes are smooth and do not exhibit patterns of gradient obfuscation. Overall, it is difficult to interpret these figures further, but they do complement the numerical analyses done so far.

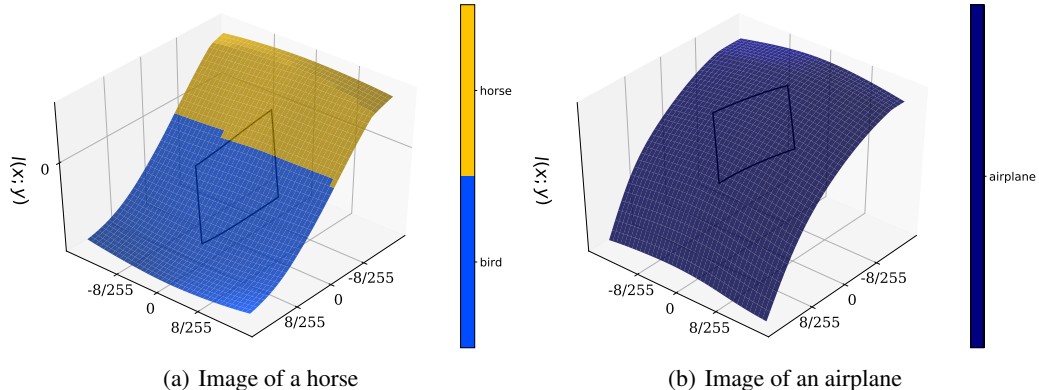

(a) Image of a horse  (b) Image of an airplane

Figure 10: Loss landscapes around the first two images from the CIFAR-10 test set for the WRN-70-16 network trained with *CutMix*. This model obtains 60.07% robust accuracy. It is generated by varying the input to the model, starting from the original input image toward either the worst attack found using $\text{PGD}^{40}$ ($u$ direction) or a random Rademacher direction ($v$ direction). The loss used for these plots is the margin loss $z_y - \max_{i \neq y} z_i$ (i.e., a misclassification occurs when this value falls below zero). The diamond-shape represents the projected $\ell_\infty$ ball of size $\epsilon = 8/255$ around the nominal image.

# B    Analysis of augmentations

In this section, we give the implementation details of the augmentations used in the paper and we complete our study with two additional augmentations: AutoAugment [12] and RandAugment [13]. Finally, we perform an ablation study analyzing the robust performance of each of the individual augmentations which compose RandAugment and we propose a curated version of RandAugment based on this ablation study.

**Implementation details of the augmentations used in the paper.**    For *Pad & Crop*, we first pad the image by 4 pixels on each side and then take a random $32 \times 32$ crop. For *MixUp* [62], we sample the image mixing weight with a beta distribution $\text{Beta}(\alpha, \alpha)$ with $\alpha = 0.2$ in our default case. For *Cutout* [15], we use a square window of size 16 whose center is uniformly sampled within the image bounds (but the window can overflow from the image). The square window is filled with the mean pixel values of the dataset. For *CutMix* [58], we use a square window whose length is randomly sampled such that the patch area ratio follows a beta distribution $\text{Beta}(\alpha, \beta)$ with parameters $\alpha = 1$ and $\beta = 1$ and whose center is uniformly sampled within the image bounds (but the window can overflow from the image). For *RICAP* [50], the size of the four patches is determined by the image center point whose coordinates are sampled with a beta distribution $\text{Beta}(\alpha, \beta)$ with parameters $\alpha = 0.3$ and $\beta = 0.3$.

| AUGMENTATIONS | M=3 | M=5 |
|---|---|---|
| AutoContrast | -0.20% | 0.20% |
| Equalize | 0.78% | 0.59% |
| Invert | **-6.84%** | **-5.18%** |
| Rotate | 0.20% | -0.39% |
| Posterize | -0.59% | **-4.20%** |
| Solarize | **-2.15%** | 0.49% |
| Color | 0.39% | 0.68% |
| Contrast | 1.17% | 0.20% |
| Brightness | 0.98% | 0.59% |
| Sharpness | -0.39% | 0.10% |
| ShearX | 0.00% | -0.68% |
| ShearY | -0.59% | -0.10% |
| TranslateX | 0.78% | -0.39% |
| TranslateY | 0.10% | 0.00% |
| SolarizeAdd | 0.00% | -0.20% |

Table 5: Delta in robust accuracy on the validation split of CIFAR-10 for each augmentation compared to the median robust accuracy over all the tested augmentations. The median robust accuracies are 53.61% and 53.81% for $M = 3$ and $M = 5$, respectively. We highlight in bold the results where the robust accuracy is significantly lower than the median robust accuracy.

Regarding how data augmentation should be inserted in the adversarial training pipeline, there are two possible designs: applying the augmentation before or after the adversarial attack. Experimentally, we did find that applying the augmentation prior to running the attack was significantly better. Indeed, applying the augmentation after the attack significantly reduces the strength of the attack, since composition technique will destroy adversarial perturbations.

Regarding the training loss, we adapt the TRADES objective to the data augmentation setting leading us to minimize the following loss: $l_{ce}(f(\mathbf{x}';\theta), \mathbf{y}') + \beta\mathbf{D}_{KL}(\mathbf{f}(\mathbf{x}' + \delta';\theta), \mathbf{f}(\mathbf{x}';\theta))$ where $\delta' = \text{argmax}_{\delta \in S}D_{KL}(f(\mathbf{x}' + \delta; \theta), \mathbf{y}')$ with $\mathbf{x}'$ and $y'$ the result of various data augmentations. This formulation is inspired from Gowal et al. [20] where they explored various schemes for the inner optimization (see TRADES-XENT in [20]).

**Additional augmentations: AutoAugment and RandAugment.** *AutoAugment* [12] learns augmentation policies from data by using a RNN controller and a Proximal Policy Optimization algorithm. The learned policies available online have been tuned to optimize the natural accuracy and might not suit the adversarial setting. We evaluate in the adversarial setting these nominally trained policies by training a WRN-28-10 on CIFAR-10. By combining *Pad & Crop* and *AutoAugment*, our model with WA reaches 55.68% robust accuracy (and 86.51% clean accuracy). Hence, the robust performance of *AutoAugment* is similar to the one of *MixUp* (i.e. 55.62% robust accuracy) and it is performing much worse than *CutMix* which gets 57.50% robust accuracy.

*RandAugment* [13], based on *AutoAugment*, has similar performance in the standard training setting but it has a much smaller search space as it requires only two search sweeps: one over the length of the transformations sequence $N$ and a second over the global transformation magnitude $M$. This allows to quickly adapt *RandAugment* to the adversarial setting compared to *AutoAugment* and its computationally expensive search algorithm. We based our implementation of *RandAugment* on the version found in the *timm* library [55] as in the original code of *RandAugment* a small global magnitude $M$ can produce high magnitude transformations for some augmentations like *Posterize*. After running a sweep over $N = \{1, 2, 3\}$ and $M = \{1, 3, 5, 7\}$ when training adversarially a WRN-28-10 on CIFAR-10, the best model using *RandAugment* with WA reaches 55.46% robust accuracy against AA+MT (and 86.65% clean accuracy), which a robust performance similar to the one obtained with *AutoAugment*.

Finally, we noted in the main paper that, in the context of adversarial training, some augmentation techniques can perform worse than others such as *MixUp* which leads to underfitting. So, we should study separately how each of the *RandAugment* individual transformations performs for adversarial robustness. For each studied augmentation, we proceed as follows: (1) restrict the pool of available operations to the studied augmentation, (2) run RandAugment with $N = 1$ and $M = \{3, 5\}$ to train a

WRN-28-10 on CIFAR-10. To compare the augmentations against each other, we evaluate the trained models on the validation split, and report in Table 5 the delta in robust accuracy compared to the median robust accuracy over all the tested transformations. We note that some augmentations, namely *Invert*, *Posterize* and *Solarize*, significantly hurt robustness compared to the other augmentations. Hence, we run a curated version of RandAugment without these operations. After running a sweep over $N = \{1, 2, 3\}$ and $M = \{1, 3, 5, 7\}$ when training adversarially a WRN-28-10 on CIFAR-10, the best model using this curated *RandAugment* with WA reaches 56.51% robust accuracy against AA+MT (and 86.30% clean accuracy). Hence, the curated version of *RandAugment* leads to an improvement of +1.05% in robust accuracy compared to the non-curated version but is still worse than *CutMix*. Nevertheless, *RandAugment* is complementary to spatial composition techniques such as *CutMix*. So, when combining the curated *RandAugment* with *CutMix* and re-running the same search sweeps, the best model with WA reaches 58.09% robust accuracy against AA+MT, an improvement of +0.59% over *CutMix* alone with WA.

## C   Additional data setting

For completeness, we also evaluate our models in the setting that considers additional external data extracted from 80M-TI [7]. In this setting (with 500K added images), we observe in Table 6 that *CutMix* performs worse than the baseline when the model is small (i.e., WRN-28-10). This is expected as the external data should generally be more useful than the augmented data and the model capacity is too low to take advantage of all this additional data. However, *CutMix* is beneficial in the setting with external data when the model is large (i.e., WRN-70-16). It improves upon the current state-of-the-art set by Gowal et al. [20] by +0.68% (in the $\ell_\infty$ setting) and +1.19% (in the $\ell_2$ setting) in robust accuracy, thus leading to a new state-of-the-art robust accuracy of respectively 66.56% and 82.23% when using external data.

| SETUP | $\ell_\infty$ | | $\ell_2$ | |
|---|---|---|---|---|
| | CLEAN | ROBUST | CLEAN | ROBUST |
| **WRN-28-10** | | | | |
| Gowal et al. [20] (trained by us) | 89.42% | **63.05%** | 94.01% | 80.08% |
| Ours (CutMix) | 89.90% | 62.06% | 94.96% | **80.96%** |
| **WRN-70-16** | | | | |
| Gowal et al. [20] (trained by us) | 90.51% | 65.88% | 94.19% | 81.04% |
| Ours (CutMix) | 92.23% | **66.56%** | 95.74% | **82.23%** |

Table 6: Clean (without adversarial attacks) accuracy and robust accuracy (against AA+MT) on CIFAR-10 as we both test against $\epsilon_\infty = 8/255$ and $\epsilon_2 = 128/255$ in the setting with added images from 80M-TI.

## D   Societal impact

Improving robustness can have a negative impact for various reasons: (1) one can create stronger adversarial images by maximizing the error of the improved model as done in the image synthesis paper by Santurkar et al. [46], (2) the increased influence of the training points on the model when using adversarial training can reduce the sensitivity of the model and lead to larger biases [63], (3) Song et al. [47] show that improved robustness increases the success of privacy attacks and that robust models are more sensitive to membership inference attacks.