# OpenReview forum: "Data Augmentation Can Improve Robustness"
_NeurIPS.cc/2021/Conference — NeurIPS 2021 Poster_

### Official Review · Reviewer_NzdM · 2021-06-28

**Rating:** 7
**Confidence:** 4

**Summary:**

The authors combine adversarial training with popular data augmentation techniques. They discuss how model averaging and data augmentation can be combined to prevent robust overfitting. The authors evaluate their method on several datasets and show state-of-the-art results on CIFAR10.

**Limitations And Societal Impact:**

Yes, the authors discussed the limitations and potential negative societal impacts of their work.

**Main Review:**

### Quick justification of rating:
I enjoyed reading this paper and think the authors did a good job first explaining, isolating and motivating the problem, then reviewing related work and possible solutions and finally, presenting their own solution along with the necessary experiments and ablation studies.

I second the authors’ sentiment in line 76: “As such, it is rather surprising that they remain ineffective when training adversarially robust networks [15, 38, 49].” Data augmentation techniques are popular in e.g. domain adaptation and corruption robustness. It is an interesting idea to try to understand under which conditions data augmentation can be beneficial for adversarial robustness. Therefore, I think this paper will be relevant to the adversarial community.

The authors report a new state of the art on CIFAR10 for adversarial robustness with their method.

I would appreciate it if the authors answered my two comments below in the rebuttal and also clarified them in the main paper.

1) Figure 6c: Model selection. The authors show here how robust accuracy changes depending on the window length. 1) Did the authors then use the determined best value for the window length (20) to report their main results in Table 3? 2) Was this hyperparameter chosen on a separate validation set or on the CIFAR10 test set? If it was chosen on the test set, is this model selection process legit in this case? (Hyperparameters should be chosen on a separate validation set.) 3) Was the same value used to obtain the results on the other datasets in Table 2? The same questions apply to all other used data augmentation techniques, e.g. which values were used for Cutout. Maybe, it would be helpful if the authors considered subscripting the hyperparameter for the used technique, e.g. writing Cutmix_{20} if this value is different between datasets.

2) No adaptive attacks were considered. Please see [1] and [2] for why adaptive attacks should always be performed when proposing a new defense.

### Paper writing: very good

The paper is very well written, easy to follow, and I really enjoyed reading it. The authors motivate their method well and also provide a good context for their paper in the related work section. The methodology in section 3 is explained well, e.g. all variables are explained, the definitions are precise. The concept of “robust overfitting” and possible solutions are explained well and are easy to understand.

I greatly appreciate the statement in Fig. 1: “Our method builds on Gowal et al. [15] (shown above with 57.20%) and explores how augmented data can be used to improve robust accuracy by +2.87% without using any additional external data.” This statement allows to easily place this paper in context with related work.

### General comments:

Line 114: Model Averaging is a popular technique in teacher-student training in domain-adaptation where the teacher model uses the EMA weights of the student model, e.g. [1]. I wonder whether there is a connection / interpretation here, similar to how MA is used in domain adaptation?

Line 150: minor comment: Writing “verifying the hypothesis” reads like the authors pursue “an agenda” and design the experiment to confirm their hypothesis. I would suggest rephrasing the statement as “Testing the hypothesis”.

Paragraph in line 150: I think it would be important to explain how “Mixup” and “Cutmix” are implemented. For Mixup, are the authors interpolating between clean and adversarial images? Same for Cutmix, are authors replacing a part of a clean image with an adversarial image? Does it make a difference how the techniques are applied? E.g. does it make a difference for Mixup whether the mixing images are of the same kind, e.g. we interpolate only between clean images and only between adversarial images?

Fig3b: Could the authors explain the drop in robust accuracy after the learning rate drop with WA?

Line 196: “pick the best model by evaluating the robust accuracy on the same validation set.” I realize that the authors follow the procedure in Gowal et al, but this choice seems strange to me. In [2], the authors argue that in adversarial robustness, we are interested in the worst-case robustness and this is the value that should be reported. I am therefore surprised by the unintuitive choice to pick the best model here.
[2] Nicholas Carlini, Anish Athalye, Nicolas Papernot, Wieland Brendel, Jonas Rauber, Dimitris Tsipras, Ian Goodfellow, Aleksander Madry, Alexey Kurakin, “On evaluating adversarial robustness”

I like Fig. 4 since it shows clearly that using WA is better than not using WA and the color code makes it easy to quickly find the different models.

Line 221: lesser -> worse

Figure 6a: Please include the number for alpha=0. Similar for 6b and 6c, please include values for not using the data augmentation techniques for comparison.

Section 6.3. I found this section a bit hard to understand. I would suggest to the authors to look into [3] where the authors quantify the error consistency between different DNNs (and humans). Maybe the metrics used in [3] would be valuable for this study, and easier to understand than the current setting.
[3] Robert Geirhos, Kristof Meding, Felix A. Wichmann, „Beyond accuracy: quantifying trial-by-trial behaviour of CNNs and humans by measuring error consistency”

Figure 7 is mentioned before Figure 8. The ablation mentioned in Figure 7 does not really fit into subsection 6.3.

Table 2 is mentioned after Table 3, please fix this.

The authors should write what they have put into the Supplement. Currently, the supplement is not even mentioned in the paper. Please include a paragraph detailing the additional experiments and studies.

I appreciate showing results for black-box attacks in Table 4 in the Supplement. Could the authors compare to Gowal et al here as well as they have done in all other evaluations?

[1] Tarvainen and Valpola: “Mean teachers are better role models: Weight-averaged consistency targets improve semi-supervised deep learning results”
[2] Florian Tramer, Nicholas Carlini, Wieland Brendel, Aleksander Madry: "On adaptive attacks to adversarial example defenses"

### Review updates post rebuttal

I read the other reviews and the authors' responses.

I think the authors did a good job addressing my comments in their rebuttal. In particular, I had two main comments about hyperparameter selection and adaptive attacks. Considering my first comment: The authors wrote in their rebuttal that they used default hyperparameters from the papers where the augmentation techniques were introduced. I think this is a justified choice which means that they definitely did not "cheat" during model selection by choosing hyperparameters on the test set. Additionally, this choice makes their models comparable to models trained by other researchers trained with the used techniques when using the default hyperparameters. Considering adaptive attacks, the authors have a point that those attacks are primarily needed for cases when gradient obfuscation is a concern or when customized elaborate defenses are implemented which make the use of standard attacks impossible. Since they use standard training here and merely add data augmentation and EMA of the weights, gradient obfuscation isn't a concern.

I think this submission is publication-worthy because:
1.	Data augmentation is the standard "go-to" tool to increase robustness towards common corruptions. In self-supervised learning, for example in contrastive learning, choosing the appropriate data augmentation is key. It is peculiar that data augmentation (being such a vital ingredient in other areas of deep learning) has not worked at all so far for adversarial robustness. The authors are the first ones to show what needs to be done to make data augmentation work for adversarial robustness. This is (for me) the main novelty of this paper: The authors take a very popular technique that hasn't worked at all so far and make it work.
2.	The authors report a new state of the art in adversarial robustness on CIFAR10.

Thus, I still think it is a good paper and should be accepted.

**Time Spent Reviewing:**

3

---

> ### Author Response · Authors · 2021-08-09
> **Thank you for the review**
>
> Thank you for the thoughtful review. See answers below.
>
> > I enjoyed reading this paper and think the authors did a good job first explaining, isolating and motivating the problem, then reviewing related work and possible solutions and finally, presenting their own solution along with the necessary experiments and ablation studies.
>
> We really appreciate such comments.
>
> > Figure 6c: Model selection. The authors show here how robust accuracy changes depending on the window length. 1) Did the authors then use the determined best value for the window length (20) to report their main results in Table 3?
>
> No. For all results (except those reported in Fig 6(c)), we used the default settings proposed in the CutMix paper for CIFAR-10. In particular, the window length is randomly sampled such that the patch area ratio follows a beta distribution with parameters $\alpha = 1$ and $\beta = 1$. This is now clarified in the text.
>
> > 2) Was this hyperparameter chosen on a separate validation set or on the CIFAR10 test set? If it was chosen on the test set, is this model selection process legit in this case? (Hyperparameters should be chosen on a separate validation set.)
>
> We fully understand the concern. As stated above, except for specific ablation experiments, all hyper-parameters related to the various augmentation techniques and weight averaging were determined a priori (either using the default proposed in the corresponding paper, e.g., $\alpha = 0.2$ for MixUp or values adapted to our longer training schedule, e.g., EMA decay of 0.999 for weight averaging).
>
> > 3) Was the same value used to obtain the results on the other datasets in Table 2? The same questions apply to all other used data augmentation techniques, e.g. which values were used for Cutout. Maybe, it would be helpful if the authors considered subscripting the hyperparameter for the used technique, e.g. writing Cutmix_{20} if this value is different between datasets.
>
> This information is indeed missing and has been added to the text. We used $\alpha = 0.2$ for MixUp, $\alpha = 1$ and $\beta = 1$ for CutMix, window length of 16 for CutOut, padding of 4 for Pad&Crop. For all other augmentations, we used the default settings adapted to CIFAR-10.
>
> > No adaptive attacks were considered. Please see [1] and [2] for why adaptive attacks should always be performed when proposing a new defense.
>
> The defense proposed uses standard TRADES/adversarial training and does not try to mask gradients (there is no test-time modifications). We understand the need for adaptive attacks, however, in this case, we believe that strong white-box and black-box attacks are adapted. Please see the various checks performed in Appendix A.
>
> > Line 114: Model Averaging is a popular technique in teacher-student training in domain-adaptation where the teacher model uses the EMA weights of the student model, e.g. [1]. I wonder whether there is a connection / interpretation here, similar to how MA is used in domain adaptation?
>
> There is definitely a connection. We believe that weight averaging plays a crucial role in our setup due to its emulation of ensembling. See Section 6.3 for more details.
>
> > Line 150: minor comment: Writing “verifying the hypothesis” reads like the authors pursue “an agenda” and design the experiment to confirm their hypothesis. I would suggest rephrasing the statement as “Testing the hypothesis”.
>
> Thank you for the suggestion. The proposed text is better and has been changed in the manuscript.
>
> > Paragraph in line 150: I think it would be important to explain how “Mixup” and “Cutmix” are implemented. For Mixup, are the authors interpolating between clean and adversarial images? Same for Cutmix, are authors replacing a part of a clean image with an adversarial image? Does it make a difference how the techniques are applied? E.g. does it make a difference for Mixup whether the mixing images are of the same kind, e.g. we interpolate only between clean images and only between adversarial images?
>
> We realized that we forgot to include these details. They are now added in a new Appendix B. In particular, we minimize the following loss: $l_\textrm{ce}(f({\bf x}'; \theta), y') + \beta D_\textrm{KL}(f({\bf x}' + \delta'; \theta), f({\bf x}'; \theta))$ where $\delta' = \textrm{argmax}_{\delta \in S} D_\textrm{KL}(f({\bf x}' + \delta; \theta), y')$ with $\bf{x}'$ and $y'$ the result of various data augmentations. Note that our training code is available online and will be linked within the paper.
>
> > Fig3b: Could the authors explain the drop in robust accuracy after the learning rate drop with WA?
>
> The drop in robust accuracy stems from the fact that when the learning rate changes, the models that are being averaged by running the exponential moving averaging on the weights are wildly different. This stabilizes as older models are "forgotten".
>
> > Line 196: “pick the best model by evaluating the robust accuracy on the same validation set.” I realize that the authors follow the procedure in Gowal et al, but this choice seems strange to me. In [2], the authors argue that in adversarial robustness, we are interested in the worst-case robustness and this is the value that should be reported. I am therefore surprised by the unintuitive choice to pick the best model here.
>
> This model selection process is perfectly valid since it relies on the performance over a separate validation set. Furthermore, over 10 separate runs, the average difference in robust accuracy between the best and worse model is only 0.12%. All comparisons are done using the same selection strategy.
>
> > I like Fig. 4 since it shows clearly that using WA is better than not using WA and the color code makes it easy to quickly find the different models.
>
> Thank you.
>
> > Line 221: lesser -> worse
>
> Thank you. This is now corrected.
>
> > Figure 6a: Please include the number for alpha=0. Similar for 6b and 6c, please include values for not using the data augmentation techniques for comparison.
>
> Without any Pad&Crop the models with WA and without WA achieve 49.74% and 42.27%, respectively. With Pad&Crop (which is included by default), the models with WA and without WA achieve 54.44% and 53.66%. The caption now includes these results.
>
> > Section 6.3. I found this section a bit hard to understand. I would suggest to the authors to look into [3] where the authors quantify the error consistency between different DNNs (and humans). Maybe the metrics used in [3] would be valuable for this study, and easier to understand than the current setting.
>
> Section 6.3 demonstrates that the sequence of models (various snapshots throughout training) trained with CutMix tends to be diverse (less overlap between examples that are attacked) and that weight averaging effectively emulates ensembling. We will rephrase this section and hope that the new text will be clearer.
>
> > Figure 7 is mentioned before Figure 8. The ablation mentioned in Figure 7 does not really fit into subsection 6.3.
>
> Section 6.3 tries to understand the effect of weight averaging. As such, we still think that Figure 7 fits in that section.
>
> > Table 2 is mentioned after Table 3, please fix this.
>
> We apologize. Due to space restrictions and the specificities of LaTeX, we could only fit Table 2 before Table 3.
>
> > The authors should write what they have put into the Supplement. Currently, the supplement is not even mentioned in the paper. Please include a paragraph detailing the additional experiments and studies.
>
> We have now added that information in the main manuscript.
>
> > I appreciate showing results for black-box attacks in Table 4 in the Supplement. Could the authors compare to Gowal et al here as well as they have done in all other evaluations?
>
> After discussion with the authors from Gowal et al., we received their AutoAttack logs and will add their results from their WRN-70-16 model.

---

### Official Review · Reviewer_Bm6o · 2021-07-15

**Rating:** 5
**Confidence:** 5

**Summary:**

This paper has demonstrated that data augmentation can be one key factor for preventing robust overfitting. This is an interesting observation as it contradicts the previous studies. The author further combined augmentations with weight averaging to improve the robustness. The experiments are extensive and show impressive results.

**Ethical Concerns:**

No ethical concerns.

**Limitations And Societal Impact:**

Yes, they have discussed the limitations and societal impacts.

**Main Review:**

----------------------------------------------------------------------
**Pros**
1. **The finding is interesting that data augmentation can prevent robust overfitting**

2. **The experiments are extensive and show significant improvement**

3. **The writing and presentation are clear**
----------------------------------------------------------------------
**Cons**
1. **Details are missing: cannot find the training objective or algorithm table**\
Did the author utilize CutMix after generating the adversarial example, or first apply CutMix then generate the adversarial example; an ablation study is needed (which one is better and why?)\
How did the author modify the TRADES [8] objective with CutMix...? I believe this part is not trivial as it contains KL divergence in the loss formula.

2. **Concerns about the analysis of data augmentation(DA): Why is CutMix the most effective method**?\
Is it related to the clean accuracy of standard ERM training?\
If so, did the author have tried AutoAugment [4] or RandAugment [5] (which shows the state-of-the-arts performance)? Since these techniques do not need other instances to define the augmentation (unlike CutMix), hence, it is more simple to analyze

3. **Concerns about the analysis of data augmentation(DA):  Which is more critical for improving the robustness, DA, or label mixing**?\
Since the mixing methods are a combination of augmentation and label mixing [3], maybe decomposing the effect might help for the analysis.\
E.g., in Figure 6, the CutMix is even effective under a window length of 30, which is extremely interesting. The window length of 30 will not contain an important foreground feature of another instance. This result might indicate the label mixing (or smoothing) is essential.

4. **Concerns about the novelty**\
The method itself is not specified for adversarial robustness, where it can be applied to various classification problems.\
Also, It seems that weight averaging (WA) and DA is an orthogonal contribution, as the DA prevents the overfitting and WA is for improving the generalization. Note that [2,3] already has shown that loss landscape smoothing is effective for robust overfitting and generalization (as mentioned in line 118).

----------------------------------------------------------------------
**Questions**
1. Is it necessary to use WA rather than Sharpness-aware-minimization (SAM) [6]? Since SAM is known to be an advanced version of WA, it might show better performance. Note that SAM and AWP [1] share the same idea.

2. The author used Adam optimizer for the inner maximization, which is not common (I only have seen such optimization in local linear regularization [7]). Is there a reason that the authors used Adam rather than the common PGD?

3. Why is the combination of AutoAttack and Multi-target attack is less effective than the AutoAttack…?

4. In Figure 1, which model architecture is used for the comparison?

5. What does "(retrained)" indicate in Table 2? Especially for SVHN and TinyImageNet. Is it the retrained version of Cui et al., or Gowal et al.?

----------------------------------------------------------------------
**To sum up, I recommend (weak) rejection for this conference**.\
I believe the author needs more analysis on the data augmentation as it is the main contribution of this paper. Also, building a method that is specified for adversarial robustness will be great.

Also, some details are missing or questionable. I did not consider this as a negative factor for the initial review, however, the author needs to some parts for better clarification.

----------------------------------------------------------------------
**You can find the references here**\
[1] “Adversarial Weight Perturbation Helps Robust Generalization”, Wu et al., NeurIPS 2020\
[2] “Robust Overfitting may be mitigated by properly learned smoothening”, Chen et al., ICLR 2021\
 [3] “On the Generalization Effects of Linear Transformations in Data Augmentation”, Wu et al., ICML 2020\
[4] “AutoAugment: Learning Augmentation Strategies from Data”, Cubuk et al., CVPR 2019\
[5] “RandAugment: Practical automated data augmentation with a reduced search space”, Cubuk et al., NeurIPS 2020\
[6] “Sharpness-Aware Minimization for Efficiently Improving Generalization”, Foret et al., ICLR 2021\
 [7] “Adversarial Robustness through Local Linearization”, Qin et al., NeurIPS 2019\
[8] “Theoretically Principled Trade-off between Robustness and Accuracy”, Zhang et al., ICML 2019

**Time Spent Reviewing:**

8

---

> ### Author Response · Authors · 2021-08-09
> **Thank you for the review**
>
> Thank you for the review. See answers below.
>
> > Details are missing: cannot find the training objective or algorithm table. Did the author utilize CutMix after generating the adversarial example, or first apply CutMix then generate the adversarial example; an ablation study is needed (which one is better and why?). How did the author modify the TRADES [8] objective with CutMix...? I believe this part is not trivial as it contains KL divergence in the loss formula.
>
> We apologize for these missing details and we have now added them to the Appendix. Note that the training code is available online and we will provide a link to it.
>
> In particular, we minimize the following loss: $l_\textrm{ce}(f({\bf x}'; \theta), y') + \beta D_\textrm{KL}(f({\bf x}' + \delta'; \theta), f(\bf{x}'; \theta))$ where $\delta' = \textrm{argmax}_{\delta \in S} D_\textrm{KL}(f({\bf x}' + \delta; \theta), y')$ with $\bf{x}'$ and $y'$ the result of various data augmentations. This formulation is inspired from Gowal et al. where they explored various schemes for the inner optimization (see TRADES-XENT).
>
> Applying the augmentation after the attack significantly reduces the strength of the attack, since composition technique will destroy adversarial perturbations. As such, we did find that applying the augmentation prior to running the attack was significantly better.
>
> > Concerns about the analysis of data augmentation(DA): Why is CutMix the most effective method? Is it related to the clean accuracy of standard ERM training? If so, did the author have tried AutoAugment [4] or RandAugment [5] (which shows the state-of-the-arts performance)? Since these techniques do not need other instances to define the augmentation (unlike CutMix), hence, it is more simple to analyze
>
> We have added RandAugment and AutoAugment. Results indicate both techniques produce no improvements in robust accuracy over CutMix as they still suffer from robust overfitting (similar to Cutout). In general, we found that augmentations that focus on composition of several images worked better for adversarial training. In other words, robust accuracy and clean accuracy are not always correlated. From the methods we tried, CutMix is the only method that both prevents robust overfitting and avoids underfitting.
>
> | Augmentation | Robust (and clean) acc without WA | Robust (and clean) acc with WA |
> |---|---|---|
> | AutoAugment | 52.17% (82.91%) | 54.46% (84.71%) |
> | RandAugment | 51.20% (84.20%) | 53.30% (86.93%) |
>
> > Concerns about the analysis of data augmentation(DA): Which is more critical for improving the robustness, DA, or label mixing?
> Since the mixing methods are a combination of augmentation and label mixing [3], maybe decomposing the effect might help for the analysis. E.g., in Figure 6, the CutMix is even effective under a window length of 30, which is extremely interesting. The window length of 30 will not contain an important foreground feature of another instance. This result might indicate the label mixing (or smoothing) is essential.
>
> Doing ablation experiments on label mixing is an excellent suggestion and would be worthy of a paper in itself. Our added results on Rand/AutoAugment seem to suggest that label mixing could play an important role (to avoid robust overfitting). However, we highlight that MixUp also mixes labels but performs significantly worse than CutMix (or other composition techniques), which clearly shows that the type of image augmentations is important. We experimented with label-mixing by turning off the image augmentation within our pipeline. The label-mixing part of CutMix only achieves 54.34% robust accuracy (compared to 57.50% for full CutMix).
>
> Our work focuses on augmentation techniques that were shown to be ineffective for robust training (see Rice et al., Wu et al. and Gowal et al. who all concluded that data augmentation was not useful) and on demonstrating that combination of data augmentations with weight averaging could be leveraged to improve performance.
>
> For Fig 6 (b-c), we randomly sample the effective window length between 1 and the x-axis value. We have now clarified this in the caption.
>
> > Concerns about the novelty. The method itself is not specified for adversarial robustness, where it can be applied to various classification problems. Also, It seems that weight averaging (WA) and DA is an orthogonal contribution, as the DA prevents the overfitting and WA is for improving the generalization. Note that [2,3] already has shown that loss landscape smoothing is effective for robust overfitting and generalization (as mentioned in line 118).
>
> The main contribution of this paper is to demonstrate that WA and DA should not be disentangled and, as such, those are **not** orthogonal contributions. More importantly, it shows that DA techniques can be used to improve robustness (contrary to what has been shown in the literature so far). Finally, we also demonstrate that it is the ensembling effect of WA that helps DA techniques.
>
> We point out that [2] does not demonstrate improvement beyond of what stochastic weight averaging is capable of and [3] does not focus on robustness to lp-norm perturbations.
>
> > Is it necessary to use WA rather than Sharpness-aware-minimization (SAM) [6]? Since SAM is known to be an advanced version of WA, it might show better performance. Note that SAM and AWP [1] share the same idea.
>
> SAM/AWP and WA work in different ways and could even be combined. WA is more akin to a temporal ensemble which will benefit from diverse checkpoints. Additionally, SAM/AWP are more costly techniques since they require taking at least one additional backward pass. So, if they achieve similar performance to WA, we would rather use WA. To compare SAM/AWP and WA, we ran SAM/AWP+CutMix on a WRN-28-10 (we tried various normalization options for the weight gradients and various step-size). The best we obtained in robust accuracy was 57.08% compared 57.50% for WA+CutMix. We did only see minor improvements when combining both SAM and WA.
>
> > The author used Adam optimizer for the inner maximization, which is not common (I only have seen such optimization in local linear regularization [7]). Is there a reason that the authors used Adam rather than the common PGD?
>
> This is a particularity of our pipeline as this setup allows to easily use different losses without the need to tune the inner optimization step-size. Note that there is no significant difference between using Adam and a fixed step-size of $\epsilon / 4$ for TRADES or adversarial training. We made that decision early on and this doesn't change any of the conclusions of the paper.
>
> > Why is the combination of AutoAttack and Multi-target attack is less effective than the AutoAttack…?
>
> As explained in Section 5, AA+MT does not include the same attacks that are present in AutoAttack. We used the same notation as Gowal et al. to avoid additional confusions. We also note that AA+MT is generally stronger than AA (except for CIFAR-100) which is the reason why we use it (see Gowal et al. and RobustBench for more details).
>
> > In Figure 1, which model architecture is used for the comparison?
>
> This is not a comparative figure. This figure simply shows the various submissions to RobustBench throughout the years. We have clarified the caption. Different comparisons are shown in Section 6.
>
> > What does "(retrained)" indicate in Table 2? Especially for SVHN and TinyImageNet. Is it the retrained version of Cui et al., or Gowal et al.?
>
> This is indeed unclear. It indicates the models that we have retrained according to Gowal et al.'s methodology. We have modified the text to reflect this.

---

> > ### Comment · Reviewer_Bm6o · 2021-08-27
> > **Response**
> >
> > I sincerely apologize for the late reply. I will leave an AC letter to inform my fault. Also, thank you for your time and effort for the response.
> >
> > ------------------------------------------------------------
> >
> > After reading the rebuttal, I recognize that the SAM [1]/AWP [2] is also effective when jointly utilized with CutMix. This result shows the same message as AWP/Chen et al., 2021 [3] where they show that the loss surface smoothness is essential*. Hence, this result can hurt the hypothesis where the ensemble was emphasized.
> >
> > I do notice that the authors are arguing that the combination of DA and WA is important. However, it is still unclear why two orthogonal combinations should be effective.
> >
> > \* Also the Swish activation also has a contribution to the loss surface smoothness.
> >
> > ------------------------------------------------------------
> >
> > I also agree with the other reviewers that this paper has shown an important observation that the DAs are also important in adversarial robustness. Also, the provided rebuttal for the data augmentation is indeed interesting. However, I still believe the significance of WA is questionable as SAM/AWP is already effective (also note that WA is already used by Chen et al., 2021) and the DA/WA are orthogonal components.
> >
> > ------------------------------------------------------------
> > **You can find the references here**\
> > [1] “Sharpness-Aware Minimization for Efficiently Improving Generalization”, Foret et al., ICLR 2021\
> > [2] “Adversarial Weight Perturbation Helps Robust Generalization”, Wu et al., NeurIPS 2020\
> > [3] “Robust Overfitting may be mitigated by properly learned smoothening”, Chen et al., ICLR 2021

---

> > > ### Author Response · Authors · 2021-08-27
> > > **Answer**
> > >
> > > We thank you for these additional comments. Please see answers below:
> > >
> > > >I recognize that the SAM [1]/AWP [2] is also effective when jointly utilized with CutMix. This result shows the same message as AWP/Chen et al., 2021 [3] where they show that the loss surface smoothness is essential. Hence, this result can hurt the hypothesis where the ensemble was emphasized.
> > >
> > > Results with AWP do not impact our conclusions with respect to ensembling being effective as these are *different* methods. Our ensembling hypothesis is compatible and does not contradict the smoothness findings of AWP by Wu et al. and the paper by Chen et al. as we discuss below.
> > >
> > > Concretely, our results (given to reviewers ircd and GNo2) clearly demonstrate that ensembling helps significantly:
> > > * We observe in Figure 3.a that DA methods like CutMix or MixUp help avoid robust overfitting. Then, we show in Figure 8a that when the performance is maintained (thanks to the data augmentation), model iterations have the same global robust accuracy but also exhibit a significant variation in individual robust predictions. WA allows us to take advantage of the diversity of models seen throughout training by effectively ensembling them.
> > > * Second, as suggested by reviewer ircd, we evaluated ensembling different models trained from scratch with early stopping. This experiment shows a significant boost in performance from ensembling two early-stopped Pad&Crop models (independently trained). The boost is even stronger for two early-stopped CutMix models, thus showing that CutMix promotes more diversity than Pad&Crop and thereby better performance. This is further evidence that it is the ensembling effect of weight averaging that is mainly responsible for robustness improvements.
> > >
> > > Please see our discussions with reviewers ircd and GNo2 for extensive details about it.
> > >
> > > > I do notice that the authors are arguing that the combination of DA and WA is important. However, it is still unclear why two orthogonal combinations should be effective.
> > >
> > > Our results indicate that **DA alone** does not help (in Figure 4, CutMix without WA on a WRN-28-10 has a worse robust accuracy against AA+MT by -0.94% than Pad&Crop without WA), **WA alone** does not improve results significantly (+0.97% in robust accuracy on a WRN-28-10) and that **DA+WA** allows for very large improvement (+3.06% w.r.t. SOTA on a WRN-28-10). Hence, the combination is effective.
> > >
> > > Regarding the explanation on *why* the combination DA+WA is effective, please see the answer to the point above.
> > >
> > > >However, I still believe the significance of WA is questionable as SAM/AWP is already effective (also note that WA is already used by Chen et al., 2021) and the DA/WA are orthogonal components.
> > >
> > > The fact that SAM/AWP is effective for robustness does not put into question the significance of WA. As written in our answer to the initial review, we ran an extensive study for SAM/AWP+CutMix on a WRN-28-10 by doing a sweep over various perturbation step-sizes and trying the two normalization options of both SAM and AWP for the weight gradients. The best we obtained in robust accuracy for SAM/AWP+CutMix was 57.08% compared to 57.50% for WA+CutMix. Furthermore, SAM/AWP is more computationally expensive as it requires one additional forward/backward pass per iteration.
> > >
> > > We do not negate that SAM/AWP is a great technique as it allows, for example, ViT to outperform ResNets without pre-training (see [1]) but in this robustness setting with data augmentation, WA achieves better robust performance while being less computationally expensive. Hence, the results obtained with DA+WA are significant.
> > >
> > > Finally, the significance of WA is *not* questionable if DA/WA are orthogonal components. Indeed, we explained in the answer to the point above that DA alone does not help and WA alone brings a small improvement. So it is significant to observe that using DA and WA together leads to a strong boost in performance.
> > >
> > > [1] When Vision Transformers Outperform ResNets without Pretraining or Strong Data Augmentations, Chen et al., 2021

---

> ### Author Response · Authors · 2021-08-19
> **Completing the previous message.**
>
> We provide more extensive details to complete the previous message. Please see answers below:
>
> > If so, did the author have tried AutoAugment [4] or RandAugment [5] (which shows the state-of-the-arts performance)? Since these techniques do not need other instances to define the augmentation (unlike CutMix), hence, it is more simple to analyze.
>
> Following the results for AutoAugment and RandAugment provided above and as advised by the reviewer, we conducted a thorough analysis of RandAugment. We studied separately the performance of each of its individual transformations for adversarial robustness. For each studied augmentation, we proceed as follows: (1) restrict the pool of available operations to the studied augmentation and the identity, (2) run RandAugment with N=1 and M={1,5} where N is the length of the transformations sequence and M the transformations global magnitude. We report below the delta in robust accuracy compared to using only Pad&Crop:
>
> | Augmentation | M = 1 | M = 5 |
> |---|---|---|
> | AutoContrast | 0.48% | 0.34% |
> | Equalize | 0.38% | 0.32% |
> | Invert | -7.67% | -7.39% |
> | Rotate | -0.17% | -0.77% |
> | Posterize | -5.81% | -4.93% |
> | Solarize | -2.83% | -0.01% |
> | Color | -0.22% | -0.05% |
> | Contrast | 0.55% | 0.04% |
> | Brightness | 0.53% | 0.32% |
> | Sharpness | -0.17% | -0.20% |
> | ShearX | -0.34% | -0.31% |
> | ShearY | -1.14% | -1.30% |
> | TranslateX | 0.18% | -0.58% |
> | TranslateY | -0.75% | -1.01% |
> | SolarizeAdd | -0.27% | -2.09% |
>
> We note that some augmentations significantly hurt robustness and might explain why RandAugment achieves poor robust accuracy. We can verify this hypothesis by re-running a curated version of RandAugment without the operations which significantly harm robust accuracy, namely Invert, Posterize, Solarize, ShearY, TranslateY and SolarizeAdd. We ran a sweep over N={1,2,3} and M={1,3,5,7}, and then selected the best combination on the validation set defined in our paper. This curated RandAugment achieves 57.29% robust accuracy on a WRN-28-10 (with WA). As Pad&Crop (which includes random flips) and Cutout are applied before RandAugment (following the procedure of the RandAugment paper), this result has to be compared with the Cutout robust accuracy, which is 56.40% (with WA). Hence, the curated RandAugment improves robust accuracy by +0.89% over Cutout but is still below the 57.50% achieved by CutMix. We would like to highlight that the result achieved by the curated RandAugment is consistent with the message conveyed in our paper. Indeed, compared to the curves in Figure 3, the robust accuracy curve for the curated RandAugment without WA is less flat than the CutMix curve but much flatter than the Cutout curve and thus suffers less from robust overfitting than Cutout (but still suffers from slight robust overfitting).
>
> > E.g., in Figure 6, the CutMix is even effective under a window length of 30, which is extremely interesting. The window length of 30 will not contain an important foreground feature of another instance. This result might indicate the label mixing (or smoothing) is essential.
>
> We would like to clarify some important implementation details for Figure 6. Normally, CutMix does not use a fixed window length. Thus, for the window length ablation for CutMix in Figure 6, we decided to use the cutting masks of the official implementation of Cutout. It has to be noted that these cutting masks do not have to fit within the image bounds as only the window center is sampled within these bounds. Hence, for a window length of 30, the effective window will be smaller on average than 30 and will consequently contain important foreground features of another instance. In fact, it can be seen on Figure 6b that Cutout with window length 30 can still achieve good classification performance, thus corroborating that important features must still be present on the image. Still, we thank you for your excellent suggestion on label mixing as it is an interesting experiment to complete the analysis of CutMix (see the label mixing results in the previous answer).

---

### Official Review · Reviewer_GNo2 · 2021-07-16

**Rating:** 5
**Confidence:** 4

**Summary:**

This paper demonstrates that data augmentation with a weight average can mitigate the adversarial overfitting problem and also can boost the robustness with a large margin. Especially, the author found that cutmix and weight average work the best for adversarial training with large performance gain which is state-of-the-art.

**Limitations And Societal Impact:**

Yes

**Main Review:**

Originality:
This method is not a novel method combination of cutmix and weight average, however, demonstrates the adversarial training also can get benefit from data augmentation and weight average technique for the first time.  Moreover, the technique can also prevent the robust overfitting problem.

Quality:
The submission is technically sound. However, it is a bit unclear that whether a weight average boost the performance of robustness or data augmentation does. Also, does weight average help in mitigating overfitting problems or augmentation. Cutmix itself already did not suffer any overfitting while with weight average it can gain large performance gain. However, in pad&crop, weight average seems to mitigate the overfitting problem but it did not boost the performance that large enough.

As I understand, the main message is augmentation helps in robustness. However, the paper mainly figures out that model weight averaging helps robustness to a greater extent when robust accuracy between model iterations can be maintained which seems different content from the title and abstract.

Clarity: Submission is easy to understand. However, it was hard to follow what is the main claim of the paper as I aforementioned.

Significance: Results are interesting that most of the previous work claim that augmentation does not give any benefits to robustness which was wrong in some sense. They provide thorough experiments on different sizes of architecture and diverse datasets. However, since the proposed method is just combining existing methods and didn't demonstrate any plausible reason why those two techniques help in overfitting and performance, I am not sure this work is significant enough.

**Time Spent Reviewing:**

6

---

> ### Author Response · Authors · 2021-08-09
> **Thank you for the review**
>
> Thank you for the review. See answers below.
>
> > This method is not a novel method combination of cutmix and weight average
>
> To the best of our knowledge, this paper is the first to study this precise combination for robust training. We would appreciate any additional reference if that is not the case. We also point out that several prior work (Rice et al., Wu et al. and Gowal et al.) all conclude that data augmentations do not improve robust accuracy. In this paper, we demonstrate that they can improve robustness when combined with weight averaging.
>
> > it is a bit unclear that whether a weight average boost the performance of robustness or data augmentation does.
>
> We perform several ablation experiments to understand whether weight averaging (WA) alone, data augmentation (DA) alone or the combination of WA an DA boosts robustness. In particular, we highlight Fig 4. We observe that WA barely improves the robust accuracy of Pad&Crop, while it provides a large improvement to CutMix. Similarly, CutMix without WA has the same robust accuracy than Pad&Crop without WA. These two results clearly indicate that the combination of good data augmentations (like CutMix) and weight averaging is important. In summary, WA or DA alone do not improve robustness by a large margin.
>
> > does weight average help in mitigating overfitting problems or augmentation.
>
> As shown in Fig 2(b), WA still suffers from robust overfitting. Good data augmentations, on the other hand, can prevent robust overfitting but do not improve upon the best robust accuracy obtained through early stopping.
>
> > Cutmix itself already did not suffer any overfitting while with weight average it can gain large performance gain. However, in pad&crop, weight average seems to mitigate the overfitting problem but it did not boost the performance that large enough.
>
> This is correct and reinforces the conclusion of the paper that it is the combination of WA and DA that really improves robustness.
>
> > the main message is augmentation helps in robustness. However, the paper mainly figures out that model weight averaging helps robustness to a greater extent when robust accuracy between model iterations can be maintained which seems different content from the title and abstract.
>
> The main message is that data augmentation **can** improve robustness (contrary to prior beliefs) when used in combination with WA. The abstract is also very clear on the contributions of the paper. We would appreciate any concrete feedback on how to modify the abstract. Overall, the works from Rice et al., Gowal et al. and Wu et al. may confuse researchers into thinking that data augmentations are not useful for robust training. Our work clearly demonstrates that data augmentation schemes are helpful and useful.
>
> > since the proposed method is just combining existing methods and didn't demonstrate any plausible reason why those two techniques help in overfitting and performance
>
> We want to re-emphasize that we do demonstrate through many ablation experiments that the combination of data augmentations and weight averaging is important. We also explain and provide intuition as to why this is the case: adversarial training tends to produce models that are locally linear, hence WA becomes equivalent to ensembling and ensembling is stronger when aggregating good and diverse models (see Section 6.3).

---

> > ### Comment · Reviewer_GNo2 · 2021-08-20
> > **Thank you for your response**
> >
> > Thank you for your kind response.
> >
> > I understand the contribution of your work that this paper firstly shows data augmentation can also improve robustness when used in combination with WA.
> >
> > I have few more questions. Any authors' intuition or explanation about the following questions?
> > - How mixing images can prevent robust overfitting? CutMix and Mixup seem to maintain the performance without any WA in Figure 3a.
> > - Why weight average method is only helpful when the performance is maintained? If weight average can be interpreted as ensembling, diversity could be a more important factor to have better performance.
> > - Why CutMix is the essential augmentation to leverage the robustness? Why Mixup can not show similar effects as CutMix even though controlling the alpha (it only shows less than 1% gain in Figure 6a)? Is there any better augmentation type that can be suggested for robustness?

---

> > > ### Author Response · Authors · 2021-08-24
> > > **Thank you for the additional questions**
> > >
> > > We thank you for these pertinent questions. Please see answers below:
> > >
> > > > How mixing images can prevent robust overfitting? CutMix and Mixup seem to maintain the performance without any WA in Figure 3a.
> > >
> > > This is a key question and a theoretical explanation of why mixing images can prevent robust overfitting is worthy of a paper in itself. For example, the ICLR2021 paper “How Does Mixup Help With Robustness and Generalization?” by Zhang et al. shows that the Mixup loss (even without adversarial training) is an upper bound for the adversarial loss. Their proof mainly relies on the linear combination of images and labels. This hints that mixing both images and labels is helpful for robustness on top of adversarial training.
> > >
> > > On the empirical side, at the request of reviewer Bm6o, we showed that the label-mixing part of CutMix alone (when turning off the composition of images) only achieves 54.34% robust accuracy (compared to 57.50% for full CutMix). We also highlight that Cutout, which composes images with an empty patch (but does not mix labels), suffers from robust overfitting.
> > >
> > > Overall, our experiments confirm that mixing only images or only labels is not enough to prevent robust overfitting; mixing both is necessary.
> > >
> > > > Why weight average method is only helpful when the performance is maintained? If weight average can be interpreted as ensembling, diversity could be a more important factor to have better performance.
> > >
> > > When performance is not maintained, WA *is* still helpful but to a lesser extent. In this case, a compromise must be found between the performance boost from ensembling diverse model iterations and the performance loss from incorporating model iterations with degraded performance in the ensemble. This compromise can be observed for Pad&Crop in Figure 7. While for CutMix increasing the WA decay rate (i.e. ensembling more model iterations) always results in better robust performance, we observe that for Pad&Crop the maximum robust performance is obtained at $\tau = 0.9925$. When the WA decay rate becomes too large ($\tau>0.9925$), too many model iterations with degraded performance are incorporated in the ensemble, thus hurting the robust performance of the ensemble.
> > >
> > >
> > > Additionally, we agree that diversity is an important factor that provides better robustness. As a matter of fact, we show in Figure 8a that when the performance is maintained, model iterations have the same global robust accuracy but also exhibit a significant variation in individual robust predictions. As suggested by reviewer ircd, we evaluated ensembling different models trained from scratch with early stopping. This experiment shows a stronger boost in performance from ensembling two early-stopped CutMix models (independently trained) than with two early-stopped Pad&Crop models, thus showing that CutMix promotes more diversity than Pad&Crop and thereby better performance
> > >
> > >
> > > In conclusion, the diversity between model iterations can only compensate up to a certain point for the decrease in robust performance due to robust overfitting.
> > >
> > > > Why CutMix is the essential augmentation to leverage the robustness? Why Mixup can not show similar effects as CutMix even though controlling the alpha (it only shows less than 1% gain in Figure 6a)? Is there any better augmentation type that can be suggested for robustness?
> > >
> > > While both CutMix and Mixup prevent robust overfitting, we can observe in Figure 3a (by looking at the curve before the change of learning rate) that Mixup suffers from underfitting, as it is lower than the other curves. Mixup’s underfitting is further confirmed in Figure 5, as increasing Mixup mixing rate leads to worse performance before the change of learning rate. A possible explanation is that low-level features tend to be destroyed by Mixup (e.g., when blending images 50-50), whereas composition techniques maintain these low-level features (i.e., edges are still visible from both images). Hence, we hypothesize that augmentations designed for robustness need to preserve low-level features.
> > >
> > > We had to make a decision on which augmentations to study and focused predominantly on the data augmentations evaluated by Rice et al., Wu et al. and Gowal et al.. As suggested by reviewers Bm6o and ircd, we have now evaluated AutoAugment and RandAugment. We show that these SOTA methods for standard ERM training do not improve over Pad&Crop for adversarial training. Furthermore, we identified the transformations that are responsible for this poor performance and we show that a curated version of RandAugment can perform better than Cutout (but worse than CutMix) for adversarial robustness when combined with weight averaging (see answer to reviewer Bm6o). The fact that transformations such as Invert, Posterize or Solarize are detrimental to robustness further supports our hypothesis that augmentations designed for robustness need to preserve low-level features.
> > >
> > > Regarding a better augmentation type for robustness, it is possible to combine augmentation techniques which work well for robustness such as the curated RandAugment and CutMix. We ran a sweep over N={1,2} and M={1,3,5,7} where N is the length of the transformations sequence and M the transformations global magnitude, and then selected the best combination on the validation set defined in our paper. This combination of CutMix and the curated RandAugment achieves 58.44% robust accuracy on a WRN-28-10 (with WA), an improvement of +0.94% over CutMix alone. Overall, this improves over the previous SOTA method by Gowal et al. by +4.00%.
> > >
> > > We have now added a discussion section that explains (a) mixing only images or only labels is not enough to prevent robust overfitting, (b) the diversity between model iterations can only compensate up to a certain point for the decrease in robust performance due to robust overfitting, (c) our hypothesis that augmentations designed for robustness need to preserve low-level features, and (d) evaluating AWP/SAM as an alternative to WA (answer to reviewer Bm6o).

---

### Official Review · Reviewer_ircd · 2021-07-17

**Rating:** 7
**Confidence:** 4

**Summary:**

The main result of this paper is that some kinds of data augmentation can boost adversarial robustness when combined with model weight averaging. Prior work investigating data augmentation does not perform weight averaging, and reported that data augmentation wasn't useful. This paper reports empirical gains on common benchmarks with CutMix augmentation, and run some experiments to see where the gains in model weight averaging are coming from, the effect of unlabeled data, generalization across different architectures etc.

**Limitations And Societal Impact:**

Yes

**Main Review:**

Clarity: Overall the paper was very well written: claims were clear and well motivated. I had a few questions about the results/writing below:

l131-134: it is unclear to me why "model weight averaging helps robustness to a GREATER extent when the accuracy across iterations can be maintained". The plot only shows that even when there is no robust overfitting, weight averaging (WA) could be helpful

l161-175: How did you decide which augmentations to try to use? In particular, why do you only study cutout, cutmix and mixup compared to other things like rotations, autoaugment etc? It does seem like the kind of data augmentation to use is quite important empirically, and it would be nice to see a more substantial discussion about this in the paper.

l161-175: Relatedly, do augmentations that help standard training also help robust training (with WA)? What's a good heuristic to decide which augmentations to use in robust training vs standard training?

l194: It says that you train two models for each hyperparam setting so are the reported numbers in the table the average across these models? What's the variance like? From personal experience, SVHN at \eps = 8/255 is quite high variance across training runs so it would be good to check

l221-229: The paper tries to explain why MixUp doesn't work as well - by saying that the blended images might be very far from original images. However, this could be true even for CutMix where you literally replace parts of an image with another. It would be nice to have some sample illustrations of what the augmentations look like if you believe there is a significant qualitative difference between these kinds of augmentations that lead to the result.

l235-238: I didn't follow the argument here: how does adv training's underfitting determine which data augmentations work and which do not?

l249-257: The gains of data augmentation when using extra data are quite small- does this improve when you train the models for longer, if underfitting is the issue?

l46-48/ Sec 6.3: The empirical evidence about why data augmentation helps is not really substantiated because the current paper doesn't present results for what the model predictions look like when not using data augmentation.

Other questions:
1. What happens if we ensemble different models trained from scratch without data augmentation with early stopping? This is expensive but as per the intuition in the paper, this should also work just as well or better to improve the robust accuracy?
2. How did you make sure that the attack evaluation was not misleading? It's great that the authors use AutoAttack which tunes hyperparameters of the attack directly, but it would be good to know what other checks the authors used to make sure that the improvements are not simply because of bad tuning of the attacks. This is a real concern because the gains are generally a bit small

Significance: This paper makes a nice observation that data augmentation with WA improves robustness. This makes the story more consistent with standard training where data augmentation typically helps. However, the gains are not big enough to have a substantial empirical advantage. Also, the paper doesn't add much new insights into why some data augmentation reduces overfitting, why do some augmentations work better with standard training vs adversarial training? So it's unclear what future works could build on this paper.




Originality: The paper takes two well-studied concepts (data augmentation and model weight averaging) and puts them together. This work is also very close to the work of Rice et al. that study robust overfitting. They also reported the observation that data augmentation leads to lesser overfitting i.e. drop upon training (but same best robust accuracy). This paper builds on that observation to use weight averaging to get more gains from data augmentation.

**Time Spent Reviewing:**

3

---

> ### Author Response · Authors · 2021-08-09
> **Thank you for the review**
>
> Thank you for the lengthy reviews. Please see answers below.
>
> > l131-134: it is unclear to me why "model weight averaging helps robustness to a GREATER extent when the accuracy across iterations can be maintained". The plot only shows that even when there is no robust overfitting, weight averaging (WA) could be helpful
>
> Fig 2 shows that improvements in robust accuracy are bigger when there is no robust overfitting. This motivates our hypothesis. The rest of the paper tries to understand whether the hypothesis is correct. Fig 4 shows that augmentations that reduce robust overfitting benefit the most from weight averaging (i.e., CutMix, SmoothMix, MixUp and RICAP) as they show larger gains in robust accuracy.
>
> > l161-175: How did you decide which augmentations to try to use? In particular, why do you only study cutout, cutmix and mixup compared to other things like rotations, autoaugment etc? It does seem like the kind of data augmentation to use is quite important empirically, and it would be nice to see a more substantial discussion about this in the paper.
>
> We had to make a decision on which augmentations to study and focused predominantly on the data augmentations evaluated by Rice et al., Wu et al. and Gowal et al.. For completeness, we have now added AutoAugment and RandAugment to Fig 4. We note that both augmentations produce no improvements in robust accuracy over CutMix as they still suffer from robust overfitting (similar to Cutout). In general, we found that augmentations that focus on the composition of several images worked better for adversarial training. Results reported below for a WRN-28-10:
>
> | Augmentation | Robust (and clean) acc without WA | Robust (and clean) acc with WA |
> |---|---|---|
> | AutoAugment | 52.17% (82.91%) | 54.46% (84.71%) |
> | RandAugment | 51.20% (84.20%) | 53.30% (86.93%) |
>
> > l161-175: Relatedly, do augmentations that help standard training also help robust training (with WA)? What's a good heuristic to decide which augmentations to use in robust training vs standard training?
>
> That is a good question. In general, there seems to be no relationship between heuristics that work well for clean accuracy and those that work well for robust accuracy. In particular, RandAugment (now added) improves clean accuracy (over Pad&Crop) but have poor robust accuracy (without WA). Experimentally, it seems like composition techniques perform better for robust training.
>
> > l194: It says that you train two models for each hyperparam setting so are the reported numbers in the table the average across these models? What's the variance like? From personal experience, SVHN at \eps = 8/255 is quite high variance across training runs so it would be good to check
>
> When using TRADES, we obtain stable training on SVHN (adversarial training can be unstable). On average, over 10 runs, the standard deviation of the Pad&Crop WRN-28-10 on SVHN is 0.2%. Similarly for CIFAR-10, the standard deviation is 0.39%.
>
> > l221-229: The paper tries to explain why MixUp doesn't work as well - by saying that the blended images might be very far from original images. However, this could be true even for CutMix where you literally replace parts of an image with another. It would be nice to have some sample illustrations of what the augmentations look like if you believe there is a significant qualitative difference between these kinds of augmentations that lead to the result.
>
> We have rephrased this sentence. What we mean is that low-level features tend be destroyed by MixUp (e.g., when blending images 50-50), whereas composition techniques maintain these low-level features (i.e., edges are still visible from both images).
>
> > l235-238: I didn't follow the argument here: how does adv training's underfitting determine which data augmentations work and which do not?
>
> While MixUp can effectively reduce robust overfitting (e.g., when $\alpha = 1.4$ in Fig 5), this results in a large drop in robust accuracy. In particular, there is a trade-off between avoiding robust overfitting and robust accuracy (comparing $\alpha = 0.1$ with $\alpha = 1.4$). This trade-off is much less pronounced for composition techniques such as CutMix (see blue curve Fig 6(c)).
>
> > l249-257: The gains of data augmentation when using extra data are quite small- does this improve when you train the models for longer, if underfitting is the issue?
>
> No, training for twice the number of epochs did not improve robust accuracy, which further confirms the issue of capacity. In particular, the training loss reaches the same value (around 1.57 for the WRN-70-16 using CutMix and additional data from TinyImages-80M).
>
> > l46-48/ Sec 6.3: The empirical evidence about why data augmentation helps is not really substantiated because the current paper doesn't present results for what the model predictions look like when not using data augmentation.
>
> This comment is unclear. Several tables and plots show the difference between using the standard augmentation and more sophisticated augmentation schemes (Table 1 and 2, Fig 4, 6 and 7). Not using Pad & Crop performs significantly worse than using Pad&Crop (i.e., robust accuracy of 49.74% on a WRN-28-10 with WA compared to 54.44% with Pad&Crop).
>
> > What happens if we ensemble different models trained from scratch without data augmentation with early stopping? This is expensive but as per the intuition in the paper, this should also work just as well or better to improve the robust accuracy?
>
> Your intuition is mostly correct. While CutMix seems to produce models that are more diverse than Pad&Crop, the boost in performance from ensembling two early-stopped Pad&Crop models is still significant. In particular, ensembling two CutMix WRN-28-10 models results in 56.35% robust accuracy (85.97% clean accuracy), and ensembling 2 Pad&Crop models results in 55.69% robust accuracy (82.58% clean accuracy). We can also notice a large gap in clean accuracy. We have now added this information to the paper.
>
> > How did you make sure that the attack evaluation was not misleading? It's great that the authors use AutoAttack which tunes hyperparameters of the attack directly, but it would be good to know what other checks the authors used to make sure that the improvements are not simply because of bad tuning of the attacks. This is a real concern because the gains are generally a bit small.
>
> Appendix A contains more details and information. In particular, we verified that attacks converge, that it is possible to drive the robust accuracy to zero when $\epsilon$ increases, that black-box attacks do not reduce robust accuracy (Square [1] is part of AutoAttack) and that loss landscapes look reasonable.

---

### Author Response · Authors · 2021-08-19
**General answer**

We thank the reviewers for the time and effort spent reviewing our manuscript. The concerns raised on our initial submission have helped improve the paper. We summarize below the elements coming from the suggestions of the reviewers:

* We showed that RandAugment and AutoAugment, which are SOTA methods for standard ERM training do not improve over Pad&Crop for adversarial training. We identified the transformations that are responsible for this poor performance and we show that a curated version of RandAugment can perform better than Cutout (but worse than CutMix) for adversarial robustness when combined with weight averaging (see answer to reviewer Bm6o and ircd)
* We conducted ablation experiments on label mixing to show that label mixing is not responsible for the performance gain observed when using CutMix with weight averaging (see answer to reviewer Bm6o).
* We experimented with Sharpness-Aware Minimization (SAM) as another way to leverage augmentations that reduce robust overfitting. We observed that it also helps improve the robust performance but to a lesser extent than the simpler weight averaging used in the initial manuscript (see answer to reviewer Bm6o).
* As suggested by reviewer ircd, we evaluated ensembling different models trained from scratch with early stopping. The results concur with our manuscript conclusions as the boost in performance from ensembling two early-stopped Pad&Crop models (independently trained) is significant. This is further evidence that it is the ensembling effect of weight averaging that is mainly responsible for robustness improvements.

We believe that we have addressed all concerns so far. However, if you have time, please indicate if there are any other concerns of yours which we have not addressed and we would be pleased to clarify those points.

---

### Decision · Program_Chairs · 2021-09-27

**Decision:**

Accept (Poster)

**Comment:**

Despite the different scores, all four reviewers agree on two main points:

A) The empirical results, in particular the non-trivial robustness gains, are interesting.
B) The explanations given for the robustness gains are only preliminary and do not account for all observed phenomena (e.g., why certain augmentations work better than others).

The reviews then arrive at different scores based on how much importance they assign to each point.

Overall I agree with both points and see this as a borderline paper. The authors could have written a stronger paper by avoiding tenuous explanations and instead focusing on a thorough experimental evaluation. A paper with strong empirical results does not have to offer a comprehensive explanation for each observed phenomenon. Instead, it suffices to discuss possible hypotheses as directions for future work. If the authors want to provide explanations, they should be rigorously investigated with experiments designed specifically to test the explanations. Presenting explanations without corresponding experiments undermines the validity of the paper.

Nevertheless, the experimental results still outweigh the lack of rigor in the provided explanations for me and I recommend accepting the paper. For the final version, I suggest that the authors clearly separate experimental findings from potential explanations. The paper could benefit from de-emphasizing the latter, e.g., by moving some of the explanations to the appendix and instead moving more of the adversarial evaluation details to the main text.